# NETS: A Non-Equilibrium Transport Sampler

## Abstract

We propose an algorithm, termed the Non-Equilibrium Transport Sampler (NETS), to sample from unnormalized probability distributions. NETS can be viewed as a variant of annealed importance sampling (AIS) based on Jarzynski's equality, in which the stochastic differential equation used to perform the non-equilibrium sampling is augmented with an additional learned drift term that lowers the impact of the unbiasing weights used in AIS. We show that this drift is the minimizer of a variety of objective functions, which can all be estimated in an unbiased fashion without backpropagating through solutions of the stochastic differential equations governing the sampling. We also prove that some these objectives control the Kullback-Leibler divergence of the estimated distribution from its target. NETS is shown to be unbiased and, in addition, has a tunable diffusion coefficient which can be adjusted post-training to maximize the effective sample size. We demonstrate the efficacy of the method on standard benchmarks, high-dimensional Gaussian mixture distributions, and a model from statistical lattice field theory, for which it surpasses the performances of related work and existing baselines.

## 1 Introduction

The aim of this paper is to sample probability distributions supported on $\mathbb{R}^d$ and known only up to a normalization constant. This problem arises in a wide variety in applications, ranging from statistical physics (Faulkner & Livingstone, 2023; Wilson, 1974; Hénin et al., 2022) to Bayesian inference (Neal, 1993), and it is known to be challenging when the target distribution is not log-concave. In this situation, vanilla methods based on ergodic sampling using Markov Chain Monte Carlo (MCMC) or stochastic differential equations (SDE) such as the Langevin dynamics typically have very slow convergence rates, making them inefficient in practice.

To overcome these difficulties, a variety of more sophisticated methods have been introduced, based e.g. on importance sampling. Here we will be interested in a class of methods of this type which involve non-equilibrium sampling, by which we mean algorithms that attempt to sample from a non-stationary probability distribution. The idea is to first generate samples from a simple base distribution (e.g. a normal distribution) them push them in finite time unto samples from the target. Traditionally, this aim has been achieved by combining some transport using e.g. the Langevin dynamics with reweighing, so as to remove the bias introduced by the non-equilibrium quench. Neal's Annealed Importance Sampling (AIS) (Neal, 2001), Sequential Monte Carlo (SMC) methods (Del Moral, 1997; Doucet et al., 2001), or continuous-time variants thereof based on Jarzynski's equality (Hartmann et al., 2018) are ways to implement this idea in practice. While these methods work better than ergodic sampling in many instances, they too can fail when the variance of the un-biasing weights become too large compared to their mean: this arises when the samples end up being too far from the target distribution at the end of the non-equilibrium quench.

This problem suggests to modify the dynamics of the samples to help them evolves towards the target during the quench. Here we propose a way to achieve this by learning an *additional drift* to include in the Langevin SDE. As we will show below, there exists a drift that removes the need for the un-biasing weights altogether, and it is the minimizer of a variety of objective functions that are amenable to empirical estimation. In practice, this offers the possibility to estimate this drift using deep learning methods. We exploit this idea here, showing that it results in an unbiased sampling strategy in which importance weights can still be used to correct the samples exactly, but the variance of these weights can be made much smaller due to the additional transport.

To this end, our work makes the following **main contributions**:

- We present a sampling algorithm which combines annealed Langevin dynamics with learnable additional transport. The algorithm, which we call the Non-Equilbrium Transport Sampler (NETS), is shown to be unbiased through a generalization of the Jarzysnki equality.

- We show that the drift coefficient contributing this additional transport is the minimizer of two separate objective functions, which can be learned without backpropagating through solutions of the SDE used for sampling.

- We show that one of these objectives, a physics informed neural network (PINN) loss, is also an *off-policy* objective, meaning that it does not need samples from the target density. In addition, this objective controls the KL-divergence between the model and its target.

- The resultant samplers can be adapted *after training* by tuning the integration time-step as well as the diffusivity to improve the sample quality, which we demonstrate on high-dimensional numerical experiments below.

### 1.1 RELATED WORK

**Dynamical Measure Transport.** Most contemporary generative models for continuous data are built upon dynamical measure transport, in which samples from a base density are mapped to samples from a target density by means of solving ordinary or stochastic differential equations (ODE/SDE) whose drift coefficients are estimable. Initial attempts to do this started with (Chen et al., 2018; Grathwohl et al., 2019). Recent work built upon these ideas by recasting the challenge of estimating the drift coefficients in these dynamical equations as a problem of quadratic regression, most notably with score-based diffusion models (Ho et al., 2020; Song et al., 2020), and since then with more general frameworks (Albergo & Vanden-Eijnden, 2022; Lipman et al., 2022; Albergo et al., 2023; Liu et al., 2022; De Bortoli et al., 2021; Neklyudov et al., 2023). Importantly, these methods work because samples from the base and the target are readily available. Here, we show that analogous equations governing those systems can lead to objective functions that can be minimized with no initial data but still allow us to exploit the expressivity of models built on dynamical transport.

**Augmenting sampling with learning.** Augmenting MCMC and importance sampling procedures with transport has been an active area of research for the past decade. Early work makes use of the independence Metropolis algorithm (Hastings, 1970; Liu, 1996), in which proposals come from a transport map (Parno & Marzouk, 2018; Noé et al., 2019; Albergo et al., 2019; Gabrié et al., 2022) that are accepted or rejected based off their likelihood ratio with the target. These methods were further improved by combining them with AIS and SMC perspectives, learning incremental maps that connect a sequence of interpolating densities between the base and target (Arbel et al., 2021; Matthews et al., 2022; Midgley et al., 2023). Similar works in the high-energy physics community posit that interleaving stochastic updates within a sequence of maps can be seen as a form of non-equilbirium sampling (Caselle et al., 2022; Bonanno et al., 2024). Following the success of generative models built out of dynamical transport, there has been a surge of interest in applying these perspectives to sampling: Vargas et al. (2023); Berner et al. (2024) translate ideas from diffusion models to minimize the KL divergence between the model and the target, and Zhang & Chen (2021) as well as Behjoo & Chertkov (2024) reformulate sampling as a stochastic optimal control (SOC) problem. These approaches require backpropagating through the solution of an SDE, which is too costly in high dimensions. In addition, the methods based on SOC must start with samples from a point mass, which may be far from the target. Akhound-Sadegh et al. (2024) avoid the need to backpropagate through an SDE, but in the process introduce a bias into their objective function. An alternative perspective was also recently given in (Bruna & Han, 2024) by using denoising oracles to turn the original sampling problem into an easier one. A final line of work (Malkin et al., 2023) shows how ideas used for modeling distributions on graphs can be repurposed as tools for sampling, including with off-policy training (Sendera et al., 2024).

Vargas et al. (2024) establish an unbiased sampler with added transport called Controlled Monte Carlo Diffusions (CMCD) that is similar to ours. The main difference is how we learn the drift. In Vargas et al. (2024) an objective for this drift in gradient form is derived through the use of path integrals and Girsanov's theorem. This objective either needs backpropagating through the SDE or has to be computed with a reference measure, and is done on a fixed grid. In practice the latter can be

numerically unstable. Here, through simple manipulations of Fokker-Planck equations, we propose a variety of new objective functions for the additional drift, none of which require backpropagating through the simulation. In addition, our learning can be done in a optimize-then-discretize fashion so that the sampling can be done with arbitrary step size and time-dependent diffusion after learning the model. This gives us an adaptive knob to increase performance, which we demonstrate below. One of the objectives we propose is a Physics-Informed Neural Network (PINN) loss that has recently appeared elsewhere in the literature for sampling (Máté & Fleuret, 2023; Tian et al., 2024; Fan et al., 2024). Importantly, here we establish that this objective is valid in the context of annealed Langevin dynamics, and, moreover, that this PINN objective *directly controls* the KL divergence as well as the importance weights that emerge through the Jarzynski equality.

## 2 METHODS

### 2.1 SETUP AND NOTATIONS

We assume that the target distribution is absolutely continuous with respect to the Lebesgue measure on $R^d$, with probability density function (PDF) $\rho_1(x) = Z_1^{-1} e^{-U_1(x)}$: here $x \in \mathbb{R}^d$, $U : \mathbb{R}^d \to \mathbb{R}$ is a known energy potential, assumed twice differentiable and bounded below, and $Z_1 = \int_{\mathbb{R}^d} e^{-U_1(x)} dx < \infty$ is an unknown normalization constant, referred to as the partition function is physics and the evidence in statistics. Our aim is to generate samples from $\rho_1(x)$ so as to be able to estimate expectations with respect to this density. Additionally we wish to estimate $Z_1$.

To this end we will use a series of time-dependent potentials $U_t(x)$ which connects some simple $U_0(x)$ at $t = 0$ (e.g. $U_0(x) = \frac{1}{2}|x|^2$) to $U_1(x)$ at $t = 1$. For example we could use linear interpolation:

$$U_t(x) = (1 - t)U_0(x) + tU_1(x), \tag{1}$$

but other choices are possible as long as $U_{t=0} = U_0$, $U_{t=1} = U_1$, and $U_t(x)$ is twice differentiable in $(t, x) \in [0, 1] \times \mathbb{R}^d$, which we explore below. We assume that the time-dependent PDF associated with this potential $U_t(x)$ is normalizable for all $t \in [0, 1]$ and denote it as

$$\rho_t(x) = Z_t^{-1} e^{-U_t(x)}, \qquad Z_t = \int_{\mathbb{R}^d} e^{-U_t(x)} dx < \infty, \tag{2}$$

so that $\rho_{t=0}(x) = \rho_0(x)$ and $\rho_{t=1}(x) = \rho_1(x)$; we also assume that $\rho_0(x)$ is simple to sample (either directly or via MCMC or Langevin dynamics) and that its partition function $Z_0$ is known. To simplify the notations we also introduce the free energy

$$F_t = -\log Z_t. \tag{3}$$

Since $\partial_t F_t = -\partial_t \log \int_{R^d} e^{-U_t(x)} dx = \int_{R^d} \partial_t U_t(x) e^{-U_t(x)} dx / \int_{R^d} e^{-U_t(x)} dx$ we have the useful identity

$$\partial_t F_t = \int_{R^d} \partial_t U_t(x) \rho_t(x) dx. \tag{4}$$

### 2.2 NONEQUILIBRIUM SAMPLING WITH IMPORTANCE WEIGHTS

Annealed importance sampling uses a finite sequence of MCMC moves that satisfy detailed-balance locally in time but not globally, thereby introducing a bias that can be corrected with weights. Here we present a time-continuous variant of AIS based on Jarzynski equality that will be more useful for our purpose.

By definition of the PDF in (2), $\nabla \rho_t(x) = -\nabla U_t(x) \rho_t(x)$ and hence, for any $\varepsilon_t \geq 0$, we have

$$0 = \varepsilon_t \nabla \cdot (\nabla U_t \rho_t + \nabla \rho_t). \tag{5}$$

Since we also have

$$\partial_t \rho_t = -(\partial_t U_t - \partial_t F_t) \rho_t, \tag{6}$$

we can combine these last two equations to deduce that

$$\partial_t \rho_t = \varepsilon_t \nabla \cdot (\nabla U_t \rho_t + \nabla \rho_t) - (\partial_t U_t - \partial_t F_t) \rho_t. \tag{7}$$

The effect of the last term at the right hand-side of this equation can be accounted for by using weights. To see how, notice that if we extend the phase space to $(x, a) \in \mathbb{R}^{d+1}$ and introduce the PDF $f_t(x, a)$ solution to the Fokker-Planck equation (FPE)

$$\partial_t f_t = \varepsilon_t \nabla \cdot (\nabla U_t f_t + \nabla f_t) + \partial_t U_t \partial_a f_t, \qquad f_{t=0}(x, a) = \delta(a)\rho_0(x), \tag{8}$$

then a direct calculation using (4) (for details see Appendix 4.1) shows that

$$\rho_t(x) = \frac{\int_{\mathbb{R}} e^a f_t(x, a) da}{\int_{\mathbb{R}^{d+1}} e^a f_t(y, a) dady}. \tag{9}$$

Therefore we can use the solution to SDE associated with the FPE (8) in the extended space to estimate expectations with respect to $\rho_t(x)$:

**Proposition 1** (Jarzynski equality). *Let $(X_t, A_t)$ solve the coupled system of SDE/ODE*

$$dX_t = -\varepsilon_t \nabla U_t(X_t) dt + \sqrt{2\varepsilon_t} dW_t, \qquad X_0 \sim \rho_0, \tag{10}$$
$$dA_t = -\partial_t U_t(X_t) dt, \qquad\qquad A_0 = 0, \tag{11}$$

*where $\varepsilon_t \geq 0$ is a time-dependent diffusion coefficient and $W_t \in \mathbb{R}^d$ is the Wiener process. Then for all $t \in [0, 1]$ and any test function $h : \mathbb{R}^d \to \mathbb{R}$, we have*

$$\int_{\mathbb{R}^d} h(x)\rho_t(x) dx = \frac{\mathbb{E}[e^{A_t} h(X_t)]}{\mathbb{E}[e^{A_t}]}, \qquad Z_t/Z_0 = e^{-F_t + F_0} = \mathbb{E}[e^{A_t}], \tag{12}$$

*where the expectations are taken over the law of $(X_t, A_t)$.*

The proof of this proposition is given in Appendix 4.1 and it relies on the identity $\int_{\mathbb{R}^d} h(x)\rho_t(x) = \int_{\mathbb{R}^{1+d}} e^a h(x) f_t(x, a) dadx / \int_{\mathbb{R}^{d+1}} f_t(x, a) dadx$ which follows from (9). The second equation in (12) for the free energy $F_t$ is what is referred to as Jarzynski's equality, and was originally surmised in the context of non-equilibrium thermodynamics (Jarzynski, 1997).

**Remark 1.** *We stress that it is key to use the weights $e^{A_t}$ in (12) because $\rho_t(x)$ **is not the PDF of $X_t$ in general**. Indeed, if we denote by $\tilde{\rho}_t(x)$ the PDF of $X_t$, it satisfies the FPE*

$$\partial_t \tilde{\rho}_t = \varepsilon_t \nabla \cdot (\nabla U_t \tilde{\rho}_t + \nabla \tilde{\rho}_t), \quad \tilde{\rho}_{t=0} = \rho. \tag{13}$$

*This FPE misses the term $-(\partial_t U_t - \partial_t F_t)\rho_t$ at the right hand-side of (7), and as a result $\tilde{\rho}_t(x) \neq \rho_t(x)$ in general – intuitively, $\tilde{\rho}_t(x)$ lags behind $\rho_t(x)$ when the potential $U_t(x)$ evolves and this lag is what the weights in (12) correct for.*

It is important to realize that, while the relation (12) can be used to compute unbiased estimators of expectations, this estimator on its own can be high variance if the lag between the PDF $\tilde{\rho}_t(x)$ of $X_t$ and $\rho_t(x)$ is too pronounced. This issue can be alleviated by using resampling methods as is done in sequential Monte Carlo (Doucet et al., 2001). Here we will solve it by adding some additional drift in (10) that will compensate for this lag and reduce the effect of the weights.

### 2.3 Nonequilibrium sampling with perfect additional transport

To see how we can add a transport term to eliminate the need of the weights, let us introduce a velocity field $b_t(x) \in \mathbb{R}^d$ which at all times $t \in [0, 1]$ satisfies

$$\nabla \cdot (b_t \rho_t) = -\partial_t \rho_t. \tag{14}$$

We stress that this is an equation for $b_t(x)$ in which $\rho_t(x)$ is fixed and given by (2): In Appendix 4.3 we show how to express the solution to (14) via Feynman-Kac formula. If (14) is satisfied, then we can combine this equation with (5) and (6) to arrive at

$$\partial_t \rho_t = \varepsilon_t \nabla \cdot (\nabla U_t \rho_t + \nabla \rho_t) - \nabla \cdot (b_t \rho_t), \tag{15}$$

which is a standard FPE. Therefore the solution to the SDE associated with (15) allows us to sample $\rho_t(x)$ directly (without weights). We phrase this result as:

**Proposition 2** (Sampling with perfect additional transport.). *Let $b_t(x)$ be a solution to (14) and let $X_t^b$ satisfy the SDE*

$$dX_t^b = -\varepsilon_t \nabla U_t(X_t^b)dt + b_t(X_t^b)dt + \sqrt{2\varepsilon_t}dW_t, \qquad X_0^b \sim \rho_0, \qquad (16)$$

*where $\varepsilon_t \geq 0$ is a time-dependent diffusion coefficient and $W_t \in \mathbb{R}^d$ is the Wiener process. Then $\rho_t(x)$ is the PDF of $X_T^b$, i.e. for all $t \in [0, 1]$ and, given any test function $h : \mathbb{R}^d \to \mathbb{R}$, we have*

$$\int_{\mathbb{R}^d} h(x)\rho_t(x)dx = \mathbb{E}[h(X_t^b)], \qquad (17)$$

*where the expectation at the right-hand side is taken over the law of $(X_t^b)$.*

This proposition is proven in Appendix 4.1 and it shows that we can in principle get rid of the weights altogether by adding the drift $b_t(x)$ in the Langevin SDE. This possibility was first noted in Vaikuntanathan & Jarzynski (2008) and is also exploited in Tian et al. (2024) for deterministic dynamics (i.e. setting $\varepsilon_t = 0$ in (16)) and in Vargas et al. (2024) using the SDE (16). Of course, in practice we need to estimate this drift, and also correct for sampling errors if this drift is imperfectly learned. Let us discuss this second question first, and defer the derivation of objectives to learn $b_t(x)$ to Secs. 2.5 and 2.6. In Appendix 4.3 we show how to express the solution to (14) via Feynman-Kac formula.

## 2.4 Non-Equilibrium Transport Sampler

Let us now show that we can combine the approaches discussed in Secs. 2.2 and 2.3 to design samplers in which we use an added transport, possibly imperfect, and importance weights.

To this end, suppose that we wish to add an additional transport term $-\nabla \cdot (\hat{b}_t \rho_t)$ in (7), where $\hat{b}_t(x) \in \mathbb{R}^d$ is some given velocity that does not necessarily solve (14). Using the expression in (2) for $\rho_t(x)$, we have the identity

$$-\nabla \cdot (\hat{b}_t \rho_t) = -\nabla \cdot \hat{b}_t \rho_t + \nabla U_t \cdot \hat{b}_t \rho_t \qquad (18)$$

Therefore we can rewrite (5) equivalently as

$$\partial_t \rho_t = \varepsilon_t \nabla \cdot (\nabla U_t \rho_t + \nabla \rho_t) - \nabla \cdot (\hat{b}_t \rho_t) + (\nabla \cdot \hat{b}_t - \nabla U_t - \partial_t U_t + \partial_t F_t)\rho_t. \qquad (19)$$

We can now proceed as we did with (5) and extend state space to account for the effect of the terms $(\nabla \cdot b_t - \nabla U_t - \partial_t U_t + \partial_t F_t)\rho_t$ in this equation via weights, while having the term $-\nabla \cdot (\hat{b}_t(x)\rho_t(x))$ contribute to some additional transport. This leads us to a result originally obtained in Vaikuntanathan & Jarzynski (2008) and recently re-derived in Vargas et al. (2024)):

**Proposition 3** (Nonequilibrium Transport Sampler (NETS)). *Let $(X_t^{\hat{b}}, A_t^{\hat{b}})$ solve the coupled system of SDE/ODE*

$$dX_t^{\hat{b}} = -\varepsilon_t \nabla U_t(X_t^{\hat{b}})dt + \hat{b}_t(X_t^{\hat{b}})dt + \sqrt{2\varepsilon_t}dW_t, \qquad X_0^{\hat{b}} \sim \rho_0, \qquad (20)$$

$$dA_t^{\hat{b}} = \nabla \cdot \hat{b}_t(X_t^{\hat{b}})dt - \nabla U_t(X_t^{\hat{b}}) \cdot \hat{b}_t(X_t^{\hat{b}})dt - \partial_t U_t(X_t^{\hat{b}})dt, \qquad A_0^{\hat{b}} = 0, \qquad (21)$$

*where $\varepsilon_t \geq 0$ is a time-dependent diffusion coefficient and $W_t \in \mathbb{R}^d$ is the Wiener process. Then for all $t \in [0, 1]$ and any test function $h : \mathbb{R}^d \to \mathbb{R}$, we have*

$$\int_{\mathbb{R}^d} h(x)\rho_t(x)dx = \frac{\mathbb{E}[e^{A_t^{\hat{b}}}h(X_t^{\hat{b}})]}{\mathbb{E}[e^{A_t^{\hat{b}}}]}, \qquad Z_t/Z_0 = e^{-F_t + F_0} = \mathbb{E}[e^{A_t^{\hat{b}}}], \qquad (22)$$

*where the expectations are taken over the law of $(X_t^{\hat{b}}, A_t^{\hat{b}})$.*

A simple proof of this proposition using simple manipulations of the FPE is given in Appendix 4.1, which will allow us to write down a variety of new loss functions for learning $\hat{b}_t$; for an alternative proof using Girsanov theorem, see Vargas et al. (2024). For completeness, in Appendix 4.2 we also give a time-discretized version of Proposition 3, and in Appendix 4.4 we generalize it in two ways: to include inertia and to turn $t$ into a vector coordinate for multimarginal sampling. We also discuss the connection between NETS and the method of Vargas et al. (2024) in Appendix 4.8.

Notice that, if $\hat{b}_t(x) = 0$, the equations in Proposition 3 simply reduce to those in Proposition 1, whereas if $\hat{b}_t(x) = b_t(x)$ solves (14) we can show that

$$A_t^b = -F_t + F_0, \tag{23}$$

i.e. the weights have zero variance and give the free energy difference. Indeed, by expanding both sides of (14) and dividing them by $\rho_t(x) > 0$, this equation can equivalently be written as

$$\nabla \cdot b_t - \nabla U_t \cdot b_t = \partial_t U_t - \partial_t F_t. \tag{24}$$

As a result, when $\hat{b}_t(x) = b_t(x)$, (21) reduces to

$$dA_t^b = -\partial_t F_t dt, \qquad A_0^b = 0, \tag{25}$$

and the solution to this equation is (23). In practice, achieving zero variance of the weights by estimating $b_t(x)$ exactly is not generally possible, but having a good approximation of $b_t(x)$ can help reducing this variance dramatically, as we will illustrate below via experiments.

## 2.5 Estimating the drift $b_t(x)$ via a PINN objective

Equation (24) can be used to derive an objective for both $b_t(x)$ and $F_t$. The reason is that in this equation the unknown $\partial_t F_t$ can be viewed as factor that guarantees solvability: indeed, integrating both sides of (14) gives $0 = -\partial_t \int_{\mathbb{R}^d} U_t(x)\rho_t(x)dx + \partial_t F_t$, which, by (4), is satisfied if and only if $F_t$ is (up to a constant fixed by $F_0 = -\log Z_0$) the exact free energy (3). This offers the possibility to learn both $b_t(x)$ and $F_t$ variationally using an objective fitting the framework of physics informed neural networks (PINNs):

**Proposition 4** (PINN objective). *Given any $T \in (0, 1]$ and any PDF $\hat{\rho}_t(x) > 0$ consider the objective for $(\hat{b}, \hat{F})$ given by:*

$$L_{PINN}^T[\hat{b}, \hat{F}] = \int_0^T \int_{\mathbb{R}^d} \left| \nabla \cdot \hat{b}_t(x) - \nabla U_t(x) \cdot \hat{b}_t(x) - \partial_t U_t(x) + \partial_t \hat{F}_t \right|^2 \hat{\rho}_t(x)dxdt. \tag{26}$$

*Then $\min_{\hat{b}, \hat{F}} L_{PINN}^T[\hat{b}, \hat{F}] = 0$, and all minimizers $(b, F)$ are such that and $b_t(x)$ solves (24) and $F_t$ is the free energy (3) for all $t \in [0, T]$.*

This result is proven in Appendix 4.1: in practice, we will use $T \in (0, 1]$ for annealing but ultimately we are interested in the result when $T = 1$. Note that since the expectation over an arbitrary $\hat{\rho}_t(x)$ in (26), it can be used as an off-policy objective. It is however natural to use $\hat{\rho}_t(x) = \rho_t(x)$ since it allows us to put statistical weight in the objective precisely in the regions where we need $b_t(x)$ to transport probability mass. In either case, there is no need to backpropagate through simulation of the SDE used to produce data. We show how the expectation over $\rho_t(x)$ can be estimated without bias in Appendices 4.5 and 4.5.1 to arrive at an empirical estimator for (26) when $\hat{\rho}_t(x) = \rho_t(x)$. Note also that, while minimization of the objective (26) gives an estimate $\hat{F}_t$ of the free energy, it is not needed at sampling time when solving (20). An objective similar to (26) was also recently posited in Máté & Fleuret (2023); Tian et al. (2024) for use with deterministic flows. Here, we devise it in the context of augmenting annealed Langevin dynamics.

One advantage of the PINN objective (26) is that we know that its minimum is zero, and hence we can track its value to monitor convergence when minimizing (26) by gradient descent as we do below. Another advantage of the loss (26) is that it controls the quality of the transport as measured by the Kullback-Leibler divergence:

**Proposition 5** (KL control). *Let $\hat{\rho}_t$ be the solution to the transport equation*

$$\partial_t \hat{\rho}_t = -\nabla \cdot (\hat{b}_t \hat{\rho}_t), \qquad \hat{\rho}_{t=0} = \rho_0 \tag{27}$$

*where $\hat{b}_t(x)$ is some predefined velocity field. Then, we have*

$$D_{KL}(\hat{\rho}_{t=1}||\rho_1) \leq \sqrt{L_{PINN}^{T=1}(\hat{b}, F)}. \tag{28}$$

*where $F_t$ is the free energy. In addition, given any estimate $\hat{F}_t$ such that $\int_0^1 |\partial_t \hat{F}_t - \partial F_t|^2 dt \leq \delta$ for some $\delta \geq 0$, we have*

$$D_{KL}(\hat{\rho}_{t=1}||\rho_1) \leq \sqrt{2L_{PINN}^{T=1}(\hat{b}, \hat{F}) + 2\delta}. \tag{29}$$

This proposition is proven in Appendix 4.1. Notice that the bound (28) can be estimated by using $\partial_t F_t = \mathbb{E}[e^{A_t^{\hat{b}}} \partial_t U_t(X_t^{\hat{b}})]/\mathbb{E}[e^{A_t^{\hat{b}}}]$ in the PINN loss (26).

### 2.6 Estimating the drift $b_t(x) = \nabla\phi_t(x)$ via Action Matching (AM)

In general (14) is solved by many $b_t(x)$. One way to get a unique (up to a constant in space and time) solution to this equation is to impose that the velocity be in gradient form, i.e. set $b_t(x) = \nabla\phi_t(x)$ for some scalar-valued potential $\phi_t(x)$. If we do so, (14) can be written as $\nabla \cdot (\nabla\phi_t(x)\rho_t) = -\partial_t\rho_t$, and it is easy to see that at all times $t \in [0, 1]$ the solution to this equation minimizes over $\hat{\phi}_t$ the objective

$$\int_{\mathbb{R}^d} \left[\tfrac{1}{2}|\nabla\hat{\phi}_t(x)|^2\rho_t(x) - \hat{\phi}_t(x)\partial_t\rho_t(x)\right]dx$$
$$= \int_{\mathbb{R}^d} \left[\tfrac{1}{2}|\nabla\hat{\phi}_t(x)|^2 + (\partial_t U_t(x) - \partial_t F_t)\hat{\phi}_t(x)\right]\rho_t(x)dx. \tag{30}$$

If we use (4) to set $\partial_t F_t = \int_{\mathbb{R}^d} \partial_t U_t(x)\rho_t(x)dx$ we can use the objective at the right hand-side of (30) to learn $\phi_t(x)$ locally in time (or globally if we integrate this objective on $t \in [0, 1]$). Alternatively, we can integrate the objective at the left hand-side of (30) over $t \in [0, T]$ and use integration by parts for the term involving $\partial_t\rho_t(x)$ to arrive at:

**Proposition 6** (Action Matching objective). *Given any $T \in (0, 1]$ consider the objective for $\hat{\phi}_t(x)$:*

$$L_{AM}^T[\hat{\phi}] = \int_0^T \int_{\mathbb{R}^d} \left[\tfrac{1}{2}|\nabla\hat{\phi}_t(x)|^2 + \partial_t\hat{\phi}_t(x)\right]\rho_t(x)dxdt + \int_{\mathbb{R}^d} \left[\hat{\phi}_0(x)\rho_0(x) - \hat{\phi}_T(x)\rho_T(x)\right]dx. \tag{31}$$

*Then the minimizer $\phi_t(x)$ of (26) is unique (up to a constant) and $b_t(x) = \nabla\phi_t(x)$ satisfies (14) for all $t \in [0, T]$.*

This proposition is proven in Appendix 4.1. This objective is analogous to the loss presented in Neklyudov et al. (2023), but adapted to the sampling problem. In practice, we will use again $T \in (0, 1]$ for annealing, but ultimately we are interested in the resut at $T = 1$. Note that, unlike with the PINN objective (26), it is crucial that we use the correct $\rho_t(x)$ in the AM objective (31): that is, unlike (26), (31) cannot be turned into an off-policy objective.

**Remark 2.** *If we use $\hat{b}_t(x) = \nabla\hat{\phi}_t(x)$ in the SDEs in (20) and (21), we need to compute $\nabla \cdot \hat{b}_t(x) = \Delta\hat{\phi}_t(x)$, which is computationally costly. Fortunately, when $\varepsilon_t > 0$, the calculation of this Laplacian can be avoided by using the following alternative equation for $A_t^{\hat{b}}$:*

$$A_t^{\hat{b}} = \frac{1}{\varepsilon_t}[\hat{\phi}_t(X_t^{\hat{b}}) - \hat{\phi}_0(X_0^{\hat{b}})] - B_t, \tag{32}$$

*where*

$$dB_t = \partial_t U_t(X_t^{\hat{b}})dt + \frac{1}{\varepsilon_t}\partial_t\hat{\phi}_t(X_t^{\hat{b}}) + \frac{1}{\varepsilon_t}|\nabla\hat{\phi}_t(X_t^{\hat{b}})|^2dt + \sqrt{\frac{2}{\varepsilon_t}}\nabla\hat{\phi}_t(X_t^{\hat{b}}) \cdot dW_t. \tag{33}$$

*This equation is derived in Appendix 4.1.*

## 3 Numerical experiments

In what follows, we test the NETS method, for both the PINN objective (26) and the action matching objective (31), on standard challenging sampling benchmarks. We then study how the method scales in comparison to baselines, particularly AIS on its own, by testing it on an increasingly high dimensional Gaussian Mixture Models (GMM). Following that, we show that it has orders of magnitude better statistical efficiency as compared to AIS on its own when applied to the study of lattice field theories, even past the phase transition of these theories and in 400 dimensions (an $L = 20 \times L = 20$ lattice).

### 3.1 40-Mode Gaussian Mixture

A common benchmark for machine learning augmented samplers that originally appeared in the paper introducing Flow Annealed Importance Sampling Bootstrap (FAB) Midgley et al. (2023) is a 40-mode GMM in 2-dimensions for which the means of the mixture components span from $-40$ to $40$. The high variance and many wells make this problem challenging for re-weighting or locally updating MCMC processes. We choose as a time dependent potential $U_t(x)$ the linear interpolation (1) with $U_0$ the potential for a standard multivariate Gaussian with standard deviation scale $\sigma = 2$.

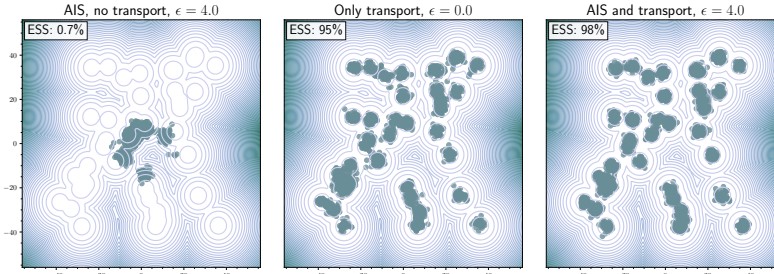

Figure 1: Comparison of the performance of annealed Langevin dynamics alone, transport alone, and annealed Langevin coupled with transport when sampling the 40-mode GMM from Midgley et al. (2023). **Left**: Annealed Langevin run for 250 steps with $\varepsilon_t = 4.0$, failing to capture the modes with $0\%$ ESS. **Center**: Learning using the PINN loss and sampling with 100 steps and $\varepsilon_t = 0$ achieves an ESS of $95\%$. **Right**: Same learning and now sampling with $\varepsilon_t = 4.0$ achieves an ESS of $98\%$.

|  | **GMM** ($d = 2$) | |
|---|---|---|
| **Algorithm** | **ESS** ↑ | $\mathcal{W}_2 \downarrow$ |
| FAB | $0.653 \pm 0.017$ | $12.0 \pm 5.73$ |
| PIS | $0.295 \pm 0.018$ | $7.64 \pm 0.92$ |
| DDS | $0.687 \pm 0.208$ | $9.31 \pm 0.82$ |
| pDEM | $0.634 \pm 0.084$ | $12.20 \pm 0.14$ |
| iDEM | $0.734 \pm 0.092$ | $7.42 \pm 3.44$ |
| CMCD-KL | $0.268 \pm 0.069$ | $9.32 \pm 0.71$ |
| CMCD-LV | $0.655 \pm 0.023$ | $4.01 \pm 0.25$ |
| NETS-AM $\varepsilon_t = 5$ (ours) | $0.808 \pm 0.031$ | $3.89 \pm 0.22$ |
| NETS-PINN $\varepsilon_t = 0$ (ours) | $\mathbf{0.954 \pm 0.003}$ | $\mathbf{3.55 \pm 0.57}$ |
| NETS-PINN $\varepsilon_t = 4$ (ours) | $\mathbf{0.979 \pm 0.002}$ | $\mathbf{3.14 \pm 0.46}$ |
| NETS-PINN-resample (ours) | $\mathbf{0.993 \pm 0.004}$ | $\mathbf{3.27 \pm 0.31}$ |

Table 1: Performance of NETS in terms of ESS and $\mathcal{W}_2$ metrics for 40-mode GMM ($d = 2$) with comparative results quoted from Akhound-Sadegh et al. (2024) for reproducibility.

We train a simple feed-forward neural network of width 256 against both the PINN objective (26), parameterizing $(\hat{b}, \hat{F})$, or the action matching objective (31), parameterizing $\hat{\phi}$. We compare the learned model from both objectives to recent related literature: FAB, Path Integral Sampler (PIS) (Zhang & Chen, 2021), Denoising Diffusion Sampler (DDS) (Vargas et al., 2023), and Denoising Energy Matching (pDEM, iDEM) (Akhound-Sadegh et al., 2024). For reproducibility with the benchmarks provided in the latter method, we compute the effective sample size (ESS) estimated from 2000 generated samples as well as

the $2-$Wasserstein ($\mathcal{W}_2$) distance between the model and the target. As noted in Table 1, all proposed variants of NETS outperform existing methods. In addition, because our method can be turned into an SMC method by including resampling during the generation, we can push the acceptance rate of the same learned PINN model to nearly $100\%$ by using a single resampling step when the ESS of the walkers dropped below $98\%$. NETS uses 100 sampling steps and an $\varepsilon_t = 0.0, 4.0$ in the SDE.

## 3.2 FUNNEL AND STUDENT-T MIXTURE

We next test NETS on Neal's funnel, a challenging synthetic target distribution which exhibits correlations at different scales across its 10 dimensions, as well as the 50-dimensional Mixture of Student-T (MoS) distribution used in Blessing et al. (2024). The definitions of the target densities and the interpolating potentials are given in Appendix 4.6. Heuristically, the first dimension is Gaussian with variance $\sigma^2 = 9$, and the other 9 dimensions are conditionally Gaussian with variance $\exp(x_0)$, creating the funnel.

We again parameterize $(\hat{b}, \hat{F})$ or $\hat{\phi}$ using simple feed forward neural networks, this time of hidden size 512. We use 100 sampling steps for both, with diffusion coefficients given in the caption of Table 2. Following Blessing et al. (2024), we compute the maximum mean discrepancy (MMD) and $\mathcal{W}_2$ distance between 2000 samples from the model and 2000 samples from the target and compare to related methods in Table 2. NETS outperforms other methods with both losses on the high dimensional MoS target in both metrics. In addition this can be improved using SMC-style resampling in the interpolation when the ESS drops below $70\%$. NETS matches the best performance in MMD for the Funnel distribution, but it is slightly worse in $\mathcal{W}_2$.

## 3.3 SCALING ON HIGH-DIMENSIONAL GMMS

In order to demonstrate that the method generalizes to high dimension, we study sampling from multimodal GMMs in higher and higher dimensions and observe how the performance scales. In

| Algorithm | Funnel ($d = 10$) | | MoS ($d = 50$) | |
|---|---|---|---|---|
| | **MMD** ↓ | $\mathcal{W}_2$ ↓ | **MMD** ↓ | $\mathcal{W}_2$ ↓ |
| FAB (Midgley et al., 2023) | **0.032 ± 0.000** | 153.894 ± 3.916 | 0.093 ± 0.014 | 1204.160 ± 147.7 |
| GMMVI (Arenz et al., 2023) | **0.031 ± 0.000** | **105.620 ± 3.472** | 0.135 ± 0.017 | 1255.216 ± 296.9 |
| PIS (Zhang & Chen, 2022) | – – | – | 0.218 ± 0.007 | 2113.172 ± 31.17 |
| DDS (Vargas et al., 2023) | 0.172 ± 0.031 | 142.890 ± 9.552 | 0.131 ± 0.001 | 2154.884 ± 3.861 |
| AFT (Arbel et al., 2021) | 0.159 ± 0.010 | 145.138 ± 6.061 | 0.395 ± 0.082 | 2648.410 ± 301.3 |
| CRAFT (Arbel et al., 2021) | 0.115 ± 0.003 | 134.335 ± 0.663 | 0.257 ± 0.024 | 1893.926 ± 117.3 |
| CMCD-KL (Vargas et al., 2024) | 0.095 ± 0.003 | 513.339 ± 192.4 | – – | – – |
| NETS-AM (ours) | 0.041 ± 0.001 | 435.793 ± 96.17 | **0.0396 ± 0.001** | **407.827 ± 69.64** |
| NETS-PINN (ours) | **0.033 ± 0.002** | 388.91 ± 141.5 | **0.032 ± 0.001** | 482.393 ± 174.6 |
| NETS-PINN-resample (ours) | **0.027 ± 0.003** | 343.78 ± 65.25 | **0.030 ± 0.000** | **400.076 ± 59.31** |

Table 2: Performance of NETS on Neal's Funnel and Mixture of Student-T distributions, measured in MMD and $\mathcal{W}_2$ distances from the true distribution. Benchmarking is in accordance with the setup of Blessing et al. (2024). Diffusion coefficient $\epsilon_t = 5, 4$ was used for NETS-AM on the Funnel and MoS, respectively. Equivalently, $\epsilon_t = 5, 5$ were used by NETS-PINN. Bold numbers are within standard deviation the best performing. Note that NETS still has perfect sample in the $\epsilon_t \to \infty$ limit, but would require finer time discretization than the 100 sampling steps used here (see Figure 4).

addition we are curious to understand how the factor in the sampling SDE coming from annealed Langevin dynamics, $\nabla U$, interacts with the learned drift $\hat{b}$ or $\nabla\hat{\phi}$ as we change the diffusivity. We construct 8-mode target GMMs in $d = 36, 64, 128, 200$ dimensions and learn $\hat{b}$ with the PINN loss in each scenario. We use the same feed forward neural network of width 512 and depth 4 to parameterize

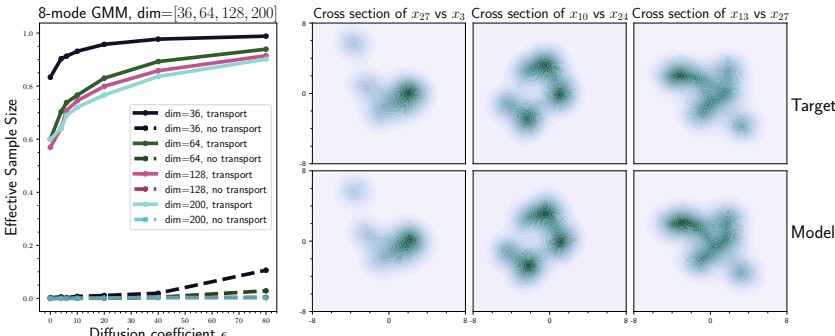

Figure 2: Demonstration of high-dimensional sampling with our method using the PINN loss in (26) and a study of how diffusivity impacts performance, with and without transport. **Left**: NETS can achieve high ESS through transport alone, and the effect of increased *diffusivity has more of a positive effect on performance with sampling than without*. AIS cannot achieve ESS above $\approx 0$ in high dimension. **Right**: Kernel density estimates of 2-$d$ cross sections of the high-dimensional, multimodal distribution arising from the model and ground truth.

both $\hat{b}$ and $\hat{F}$ for all dimensions tested and train for 4000 training iterations. Figure 2 summarizes the results. On the left plot, we note that AIS on its own cannot produce any effective samples, while even in 200 dimensions, NETS works with transport alone with 60% ESS. As we increase the diffusivity $\varepsilon_t$ and therefore the effect of the Langevin term coming from the gradient of the potential, we note all the methods converge to nearly independent sampling, and the discrepancy in performance across dimensions is diminished. Note that the caveat to achieve this is that the step size in the SDE integrator must be taken smaller to accommodate the increased diffusivity, especially for the $\varepsilon_t = 80$ data point. The number of sampling steps used to discretize the SDEs in these experiments ranged from $K = 100$ for $\varepsilon_t = 0$ up to $K = 2000$ for $\varepsilon_t = 80$. Nonetheless, it suggests that diffusion can be more helpful when there is already some successful transport than without.

### 3.4 LATTICE $\varphi^4$ THEORY

We next apply NETS to the simulation of a statistical lattice field theory at and past the phase transition from which the lattice goes from disordered, to semi-ordered, to fully ordered (neighboring

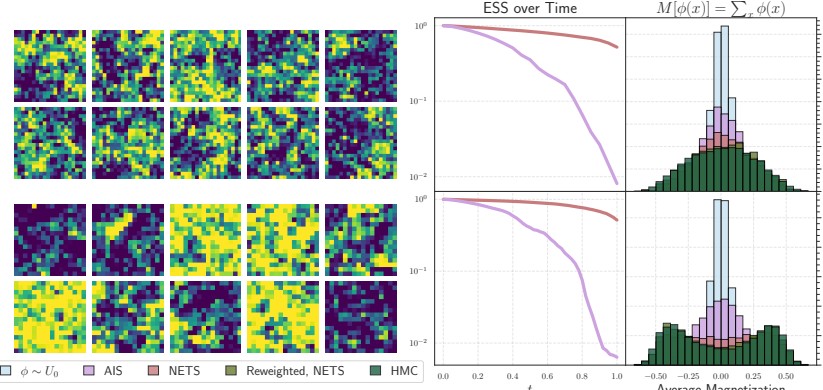

Figure 3: Comparison of the performance of NETS to AIS on two different settings for the study of $\varphi^4$ theory. **Top row, left**: 10 example generative lattice configurations with parameters $L = 20$, $m^2 = -1.0$, $\lambda = 0.9$, which demarcates the phase transition to the antiferromagnetic phase. **Top row, right**: Performance of AIS (purple curve) vs. NETS (red curve) in terms of effective sample size over time of integration $t$, and a histogram of the average magnetization of 4000 lattice configurations, sampled with AIS, NETS, and HMC (superposed in this order). Note that NETS is closer to the HMC target and re-weights correctly. Re-weighted AIS was not plotted because the weights were too high variance. **Bottom row**: Equivalent setup for $L = 16$, $m^2 = -1.0$, $\lambda = 0.8$, past the phase transition and into the ordered phase. Note that the field configurations generated by NETS are either all positive across lattice sites or all negative. AIS fails to sample the correct distribution, and its weights are too high variance to be used on the histogram.

sites are highly correlated to be of the same sign and magnitude). We study the lattice $\varphi^4$ theory in $D = 2$ spacetime dimensions. The random variables in this circumstance are field configurations $\varphi \in \mathbb{R}^{L \times L}$, where $L$ is the extent of space and time. The interpolating energy function under which we seek to sample is defined as:

$$U_t(\varphi) = \sum_x \left[ - 2 \sum_\mu \varphi_x \varphi_{x+\mu} \right] + (2D + m_t^2)\varphi_x^2 + \lambda_t \varphi_x^4, \tag{34}$$

where summation over $x$ indicates summation over the lattices sites, and $m_t^2$ and $\lambda_t$ are time dependent parameters of the theory that define the phase of the lattice (ranging from disordered to ordered, otherwise known as magnetized). A derivation of this energy function is given in Appendix 4.7. Importantly, sampling the lattice configurations becomes challenging when approaching the phase transition between the disordered and ordered phases. As an example, we identify the phase transition on $L = 16$ ($d = 256$) and $L = 20$ ($d = 400$) lattices and run NETS with the action matching loss, with $\hat{\phi}_t$ a simple feed forward neural network. We use the free theory $\lambda_0 = 0$ as the base distribution under which we initially draw samples. The definition of the target parameter values $m_1^2, \lambda_1$ both at the phase transition and in the ordered phase are given in the Appendix 4.7. In Figure 3, the top row shows samples from NETS for $L = 20$ at the phase transition, where correlations begin to appear in the lattice configurations. NETS is almost 2 orders of magnitude more statistically efficient than AIS (the same setup without the transport) in sampling at the critical point, as seen in the plot showing ESS over time. Note also that NETS can produce unbiased estimates of the magnetization as compared to a Hybrid Monte Carlo (HMC) ground truth. The bottom row shows samples past the phase transition and into the ordered phase, where the lattices begin to take on either all positive or all negative values. Again in this regime, NETS is nearly 2 orders of magnitude more statistically efficient.

While NETS performs significantly better than conventional annealed samplers on the challenging field theory problem, algorithms built out of dynamical transport still experience slowdowns near phase transitions because of the difficulty of resolving the dynamics of the integrators near these critical points. As such, we need to use 1500-2000 steps in the integrator to properly resolve the dynamics of the SDE.

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

# 4 APPENDIX

## 4.1 PROOFS OF SEC. 2

Here we provide the proofs of the statements made in Sec. 2 which, for the reader convenience, we recall.

**Proposition 1** (Jarzynski equality). *Let $(X_t, A_t)$ solve the coupled system of SDE/ODE*

$$dX_t = -\varepsilon_t \nabla U_t(X_t)dt + \sqrt{2\varepsilon_t}dW_t, \qquad X_0 \sim \rho_0, \tag{10}$$
$$dA_t = -\partial_t U_t(X_t)dt, \qquad\qquad A_0 = 0, \tag{11}$$

*where $\varepsilon_t \geq 0$ is a time-dependent diffusion coefficient and $W_t \in \mathbb{R}^d$ is the Wiener process. Then for all $t \in [0, 1]$ and any test function $h : \mathbb{R}^d \to \mathbb{R}$, we have*

$$\int_{\mathbb{R}^d} h(x)\rho_t(x)dx = \frac{\mathbb{E}[e^{A_t}h(X_t)]}{\mathbb{E}[e^{A_t}]}, \qquad Z_t/Z_0 = e^{-F_t+F_0} = \mathbb{E}[e^{A_t}], \tag{12}$$

*where the expectations are taken over the law of $(X_t, A_t)$.*

*Proof.* Let $f_t(x, a)$ with $(x, a) \in \mathbb{R}^{d+1}$ be the PDF of the joint process $(X_t, A_t)$ defined by the SDE (10) and (11). This PDF solves the FPE

$$\partial_t f_t = \varepsilon_t \nabla \cdot (\nabla U_t f_t + \nabla f_t) + \partial_t U_t \partial_a f_t, \qquad f_{t=0}(x, a) = \delta(a)\rho_0(x). \tag{35}$$

Define

$$g_t(x) = \int_{\mathbb{R}} e^a f_t(x, a)da. \tag{36}$$

We can derive an equation for $g_t(x)$ by multiplying both sides of the FPE (35) by $e^a$ and integrating over $a \in \mathbb{R}$. Using

$$\int_{\mathbb{R}} e^a \partial_t f_t(x, a)da = \partial_t \int_{\mathbb{R}} e^a f_t(x, a)da = \partial_t g_t,$$

$$\int_{\mathbb{R}} e^a \varepsilon_t \nabla \cdot (\nabla U_t f_t + \nabla f_t)da = \varepsilon_t \nabla \cdot \left( \nabla U_t \int_{\mathbb{R}} e^a f_t(x, a)da + \nabla \int_{\mathbb{R}} e^a f_t(x, a)da \right)$$
$$= \varepsilon_t \nabla \cdot (\nabla U_t g_t + \nabla g_t), \tag{37}$$

$$\int_{\mathbb{R}} e^a \partial_t U_t \partial_a f_t da = \partial_t U_t \int_{\mathbb{R}} e^a \partial_a f_t da$$
$$= -\partial_t U_t \int_{\mathbb{R}} e^a f_t da = -\partial_t U_t g_t,$$

where we arrived at the second equality in the third equation by integration by parts, we deduce that

$$\partial_t g_t = \varepsilon_t \nabla \cdot (\nabla U_t g_t + \nabla g_t) - \partial_t U_t g_t, \qquad g_{t=0}(x) = \rho_0(x) = e^{-U_0(x)+F_0}. \tag{38}$$

The solution to this parabolic PDE is unique and it can be checked by direct substitution that it is given by

$$g_t(x) = e^{-U_t(x)+F_0}. \tag{39}$$

Note that this solution is not normalized since it contains $F_0$ rather than $F_t$. In fact it is easy to see that

$$\int_{\mathbb{R}^d} g_t(x)dx = \int_{R^{d+1}} e^a f_t(x,a)dxda = e^{-F_t+F_0}, \tag{40}$$

where the first equality follows from the definition of $g_t$ and the second from its explicit expression and the definition of the free energy that implies $\int_{\mathbb{R}^d} e^{-U_t(x)}dx = e^{-F_t}$. Equation (39) is the second equation in (12). From (39) we also deduce that, given any test function $h : \mathbb{R}^d \to \mathbb{R}$, we have

$$
\begin{aligned}
\frac{\int_{R^{d+1}} e^a h(x) f_t(x,a)dxda}{\int_{R^{d+1}} e^a f_t(x,a)dxda} &= \frac{\int_{R^d} h(x)g_t(x)dx}{\int_{R^d} g_t(x)dx} \\
&= \frac{\int_{R^d} h(x)e^{-U_t(x)+F_0}dx}{\int_{R^d} e^{-U_t(x)+F_0}} \\
&= \frac{\int_{R^d} h(x)e^{-U_t(x)}dx}{\int_{R^d} e^{-U_t(x)}dx} \\
&= e^{F_t}\int_{R^d} h(x)e^{-U_t(x)}dx \\
&= \int_{R^d} h(x)\rho_t(x)dx.
\end{aligned} \tag{41}
$$

Since by definition of $f_t(x,a)$ the left hand-side of this equation can be expressed as the ratio of expectations over $(X_t, A_t)$ in the first equation in (12) we are done. $\square$

**Proposition 2** (Sampling with perfect additional transport.). *Let $b_t(x)$ be a solution to (14) and let $X_t^b$ satisfy the SDE*

$$dX_t^b = -\varepsilon_t \nabla U_t(X_t^b)dt + b_t(X_t^b)dt + \sqrt{2\varepsilon_t}dW_t, \qquad X_0^b \sim \rho_0, \tag{16}$$

*where $\varepsilon_t \geq 0$ is a time-dependent diffusion coefficient and $W_t \in \mathbb{R}^d$ is the Wiener process. Then $\rho_t(x)$ is the PDF of $X_T^b$, i.e. for all $t \in [0,1]$ and, given any test function $h : \mathbb{R}^d \to \mathbb{R}$, we have*

$$\int_{\mathbb{R}^d} h(x)\rho_t(x)dx = \mathbb{E}[h(X_t^b)], \tag{17}$$

*where the expectation at the right-hand side is taken over the law of $(X_t^b)$.*

*Proof.* If $b_t$ satisfies (14), then $\rho_t$ satisfies the FPE (15). Since (16) is the SDE associated with this FPE, (17) holds. $\square$

**Proposition 3** (Nonequilibrium Transport Sampler (NETS)). *Let $(X_t^{\hat{b}}, A_t^{\hat{b}})$ solve the coupled system of SDE/ODE*

$$dX_t^{\hat{b}} = -\varepsilon_t \nabla U_t(X_t^{\hat{b}})dt + \hat{b}_t(X_t^{\hat{b}})dt + \sqrt{2\varepsilon_t}dW_t, \qquad X_0^{\hat{b}} \sim \rho_0, \tag{20}$$

$$dA_t^{\hat{b}} = \nabla \cdot \hat{b}_t(X_t^{\hat{b}})dt - \nabla U_t(X_t^{\hat{b}}) \cdot \hat{b}_t(X_t^{\hat{b}})dt - \partial_t U_t(X_t^{\hat{b}})dt, \qquad A_0^{\hat{b}} = 0, \tag{21}$$

*where $\varepsilon_t \geq 0$ is a time-dependent diffusion coefficient and $W_t \in \mathbb{R}^d$ is the Wiener process. Then for all $t \in [0,1]$ and any test function $h : \mathbb{R}^d \to \mathbb{R}$, we have*

$$\int_{\mathbb{R}^d} h(x)\rho_t(x)dx = \frac{\mathbb{E}[e^{A_t^{\hat{b}}}h(X_t^{\hat{b}})]}{\mathbb{E}[e^{A_t^{\hat{b}}}]}, \qquad Z_t/Z_0 = e^{-F_t+F_0} = \mathbb{E}[e^{A_t^{\hat{b}}}], \tag{22}$$

*where the expectations are taken over the law of $(X_t^{\hat{b}}, A_t^{\hat{b}})$.*

*Proof.* We can follow the same steps as in the proof of Proposition 1 by considering the PDF $f_t^{\hat{b}}(x, a)$ of $(X_t^{\hat{b}}, A_t^{\hat{b}})$. This PDF solves the FPE

$$\partial_t f_t^{\hat{b}} = \varepsilon_t \nabla \cdot (\nabla U_t f_t^{\hat{b}} + \nabla f_t^{\hat{b}}) - \nabla \cdot (\hat{b}_t f_t^{\hat{b}}) - (\nabla \cdot \hat{b}_t - \nabla U_t - \partial_t U_t) \partial_a f_t^{\hat{b}},$$
$$f_{t=0}^{\hat{b}}(x, a) = \delta(a) \rho_0(x). \tag{42}$$

Define

$$g_t^{\hat{b}}(x) = \int_{\mathbb{R}} e^a f_t^{\hat{b}}(x, a) da. \tag{43}$$

We can derive an equation for $g_t^{\hat{b}}(x)$ by multiplying both sides of the FPE (42) by $e^a$ and integrating over $a \in \mathbb{R}$. Using

$$\int_{\mathbb{R}} e^a \partial_t f_t^{\hat{b}} da = \partial_t \int_{\mathbb{R}} e^a f_t^{\hat{b}} da = \partial_t g_t^{\hat{b}},$$

$$\int_{\mathbb{R}} e^a \varepsilon_t \nabla \cdot (\nabla U_t f_t^{\hat{b}} + \nabla f_t^{\hat{b}}) da = \varepsilon_t \nabla \cdot \left( \nabla U_t \int_{\mathbb{R}} e^a f_t^{\hat{b}} da + \nabla \int_{\mathbb{R}} e^a f_t^{\hat{b}} da \right),$$

$$- \int_{\mathbb{R}} e^a \nabla \cdot (\hat{b}_t f_t^{\hat{b}}) da = -\nabla \cdot \left( \hat{b}_t \int_{\mathbb{R}} e^a f_t^{\hat{b}} da \right)$$
$$= -\nabla \cdot (\hat{b}_t g_t^{\hat{b}}), \tag{44}$$

$$- \int_{\mathbb{R}} e^a (\nabla \cdot \hat{b}_t - \nabla U_t - \partial_t U_t) \partial_a f_t^{\hat{b}} da = -(\nabla \cdot \hat{b}_t - \nabla U_t - \partial_t U_t) \int_{\mathbb{R}} e^a \partial_a f_t^{\hat{b}} da$$
$$= (\nabla \cdot \hat{b}_t - \nabla U_t - \partial_t U_t) \int_{\mathbb{R}} e^a f_t^{\hat{b}} da$$
$$= (\nabla \cdot \hat{b}_t - \nabla U_t - \partial_t U_t) g_t^{\hat{b}},$$

where we arrived at the second equality in the fourth equation by integration by parts, we deduce that

$$\partial_t g_t^{\hat{b}} = \varepsilon_t \nabla \cdot (\nabla U_t g_t^{\hat{b}} + \nabla g_t^{\hat{b}}) - \nabla \cdot (\hat{b}_t g_t^{\hat{b}}) + (\nabla \cdot \hat{b}_t - \nabla U_t - \partial_t U_t) g_t^{\hat{b}},$$
$$g_{t=0}^{\hat{b}}(x) = \rho_0(x) = e^{-U_0(x) + F_0}. \tag{45}$$

The solution to this parabolic PDE is unique and it can be checked by direct substitution that it is given by

$$g_t^{\hat{b}}(x) = e^{-U_t(x) + F_0}. \tag{46}$$

This solution is not normalized since it contains $F_0$ rather than $F_t$, and it is easy to see that

$$\int_{\mathbb{R}^d} g_t^{\hat{b}}(x) dx = \int_{R^{d+1}} e^a f_t^{\hat{b}}(x, a) dx da = e^{-F_t + F_0}. \tag{47}$$

where the first equality follows from the definition of $g_t^{\hat{b}}$ and the second from its explicit expression and the definition of the free energy that implies $\int_{\mathbb{R}^d} e^{-U_t(x)} dx = e^{-F_t}$. Equation (47) is the second equation in (22). From (46) we also deduce that, given any test function $h : \mathbb{R}^d \to \mathbb{R}$, we have

$$\frac{\int_{\mathbb{R}^{d+1}} e^a h(x) f_t^{\hat{b}}(x, a) dx da}{\int_{\mathbb{R}^{d+1}} e^a f_t^{\hat{b}}(x, a) dx da} = \frac{\int_{\mathbb{R}^d} h(x) g_t^{\hat{b}}(x) dx}{\int_{\mathbb{R}^d} g_t^{\hat{b}}(x) dx}$$
$$= \frac{\int_{\mathbb{R}^d} h(x) e^{-U_t(x) + F_0} dx}{\int_{\mathbb{R}^d} e^{-U_t(x) + F_0}}$$
$$= \frac{\int_{\mathbb{R}^d} h(x) e^{-U_t(x)} dx}{\int_{\mathbb{R}^d} e^{-U_t(x)} dx} \tag{48}$$
$$= e^{F_t} \int_{\mathbb{R}^d} h(x) e^{-U_t(x)} dx$$
$$= \int_{\mathbb{R}^d} h(x) \rho_t(x) dx.$$

Since by definition of $f_t^{\hat{b}}(x, a)$ the left hand-side of this equation can be expressed as the ratio of expectations over $(X_t^{\hat{b}}, A_t^{\hat{b}})$ in the first equation in (22) we are done. $\qquad \square$

**Proposition 4** (PINN objective). *Given any $T \in (0, 1]$ and any PDF $\hat{\rho}_t(x) > 0$ consider the objective for $(\hat{b}, \hat{F})$ given by:*

$$L_{PINN}^T[\hat{b}, \hat{F}] = \int_0^T \int_{\mathbb{R}^d} \left| \nabla \cdot \hat{b}_t(x) - \nabla U_t(x) \cdot \hat{b}_t(x) - \partial_t U_t(x) + \partial_t \hat{F}_t \right|^2 \hat{\rho}_t(x) dx dt. \tag{26}$$

*Then $\min_{\hat{b}, \hat{F}} L_{PINN}^T[\hat{b}, \hat{F}] = 0$, and all minimizers $(b, F)$ are such that and $b_t(x)$ solves (24) and $F_t$ is the free energy (3) for all $t \in [0, T]$.*

*Proof.* Clearly the minimum value of (26) is zero and the minimizing pair $(\hat{b}, \hat{F})$ must satisfy

$$\nabla \cdot \hat{b}_t - \nabla U_t \cdot \hat{b}_t - \partial_t U_t + \partial_t \hat{F}_t = 0 \tag{49}$$

By multiplying both sides of this equation by $\rho_t$ is can be written as

$$\nabla \cdot (\hat{b}_t \rho_t) - \partial_t U_t \rho_t + \partial_t \hat{F}_t \rho_t = 0 \tag{50}$$

This equation requires a solvability condition obtained by integrating it over $\mathbb{R}^d$. This gives

$$-\int_{\mathbb{R}^d} \partial_t U_t(x) \rho_t(x) dx + \partial_t \hat{F}_t = 0, \tag{51}$$

which, by (4), implies that $\partial_t \hat{F}_t = \partial_t F_t$. In turn, this implies that (50) is equivalent to (14), i.e. $\hat{b}_t$ solves (14). $\square$

**Proposition 5** (KL control). *Let $\hat{\rho}_t$ be the solution to the transport equation*

$$\partial_t \hat{\rho}_t = -\nabla \cdot (\hat{b}_t \hat{\rho}_t), \qquad \hat{\rho}_{t=0} = \rho_0 \tag{27}$$

*where $\hat{b}_t(x)$ is some predefined velocity field. Then, we have*

$$D_{KL}(\hat{\rho}_{t=1} || \rho_1) \leq \sqrt{L_{PINN}^{T=1}(\hat{b}, F)}. \tag{28}$$

*where $F_t$ is the free energy. In addition, given any estimate $\hat{F}_t$ such that $\int_0^1 |\partial_t \hat{F}_t - \partial F_t|^2 dt \leq \delta$ for some $\delta \geq 0$, we have*

$$D_{KL}(\hat{\rho}_{t=1} || \rho_1) \leq \sqrt{2 L_{PINN}^{T=1}(\hat{b}, \hat{F}) + 2\delta}. \tag{29}$$

*Proof.* Consider

$$D_{\mathrm{KL}}(\hat{\rho}_t || \rho_t) = \int_{\mathbb{R}^d} \log\left(\frac{\hat{\rho}_t(x)}{\rho_t(x)}\right) \hat{\rho}_t(x) dx \tag{52}$$

where $\hat{\rho}_t$ satisfies (27). Taking the time-derivative of this expression we deduce that (using (27), $\rho_t(x) = e^{-U_t(x) + F_t}$, and multiple integrations by parts)

$$\partial_t D_{\mathrm{KL}}(\hat{\rho}_t || \rho_t) = \int_{\mathbb{R}^d} \left[ \log\left(\frac{\hat{\rho}_t(x)}{\rho_t(x)}\right) \partial_t \hat{\rho}_t(x) - \frac{\partial_t \rho_t(x)}{\rho_t(x)} \hat{\rho}_t(x) \right] dx$$

$$= \int_{\mathbb{R}^d} \left[ -\log\left(\frac{\hat{\rho}_t(x)}{\rho_t(x)}\right) \nabla \cdot (\hat{b}_t(x) \hat{\rho}_t(x)) + (\partial_t U_t(x) - \partial_t F_t) \hat{\rho}_t(x) \right] dx$$

$$= \int_{\mathbb{R}^d} \left[ \hat{b}_t(x) \cdot \nabla \log\left(\frac{\hat{\rho}_t(x)}{\rho_t(x)}\right) + \partial_t U_t - \partial_t F_t \right] \hat{\rho}_t(x) dx \tag{53}$$

$$= \int_{\mathbb{R}^d} \left[ \hat{b}_t(x) \cdot \nabla \hat{\rho}_t(x) + \left( \hat{b}_t(x) \cdot \nabla U_t(x) + \partial_t U_t - \partial_t F_t \right) \hat{\rho}_t(x) \right] dx$$

$$= \int_{\mathbb{R}^d} \left[ -\nabla \cdot \hat{b}_t(x) + \hat{b}_t(x) \cdot \nabla U_t(x) + \partial_t U_t - \partial_t F_t \right] \hat{\rho}_t(x) dx$$

Therefore

$$D_{\mathrm{KL}}(\hat{\rho}_{t=1} || \rho_1) = \int_0^1 \int_{\mathbb{R}^d} \left[ -\nabla \cdot \hat{b}_t(x) + \hat{b}_t(x) \cdot \nabla U_t(x) + \partial_t U_t - \partial_t F_t \right] \hat{\rho}_t(x) dx dt$$

$$\leq \left[ \int_0^1 \int_{\mathbb{R}^d} \left| -\nabla \cdot \hat{b}_t(x) + \hat{b}_t(x) \cdot \nabla U_t(x) + \partial_t U_t - \partial_t F_t \right|^2 \hat{\rho}_t(x) dx dt \right]^{1/2} \tag{54}$$

$$= \sqrt{L_{\mathrm{PINN}}^{T=1}(\hat{b}, F)}$$

which gives (28). To establish (29) observe that

$$L_{\text{PINN}}^1(\hat{b}, F)$$

$$= \int_0^1 \int_{\mathbb{R}^d} \left| \nabla \cdot \hat{b}_t(x) - \hat{b}_t(x) \cdot \nabla U_t(x) - \partial_t U_t + \partial_t F_t \right|^2 \hat{\rho}_t(x) dx dt$$

$$\leq 2 \int_0^1 \int_{\mathbb{R}^d} \left[ \left| \nabla \cdot \hat{b}_t(x) - \hat{b}_t(x) \cdot \nabla U_t(x) - \partial_t U_t + \partial_t \hat{F}_t \right|^2 + \left| \partial_t F_t - \partial_t \hat{F}_t \right|^2 \right] \hat{\rho}_t(x) dx dt \tag{55}$$

$$= 2 L_{\text{PINN}}^1(\hat{b}, \hat{F}) + 2 \int_0^1 |\partial_t F_t - \partial_t \hat{F}_t|^2 dt$$

Therefore, if $\int_0^1 |\partial_t \hat{F}_t - \partial_t F_t|^2 dt \leq \delta$, we have

$$L_{\text{PINN}}^1(\hat{b}, F) \leq 2 L_{\text{PINN}}^1(\hat{b}, \hat{F}) + 2\delta \tag{56}$$

Combining this bound with (28) gives (29). □

**Proposition 6** (Action Matching objective). *Given any $T \in (0, 1]$ consider the objective for $\hat{\phi}_t(x)$:*

$$L_{AM}^T[\hat{\phi}] = \int_0^T \int_{\mathbb{R}^d} \left[ \tfrac{1}{2} |\nabla \hat{\phi}_t(x)|^2 + \partial_t \hat{\phi}_t(x) \right] \rho_t(x) dx dt + \int_{\mathbb{R}^d} \left[ \hat{\phi}_0(x) \rho_0(x) - \hat{\phi}_T(x) \rho_T(x) \right] dx. \tag{31}$$

*Then the minimizer $\phi_t(x)$ of (26) is unique (up to a constant) and $b_t(x) = \nabla \phi_t(x)$ satisfies (14) for all $t \in [0, T]$.*

*Proof.* By integrating by parts in time the term involving $\partial_t \phi_t$ in the AM objective (57), we can express is as

$$L_{AM}^T[\hat{\phi}] = \int_0^T \int_{\mathbb{R}^d} \left[ \tfrac{1}{2} |\nabla \hat{\phi}_t(x)|^2 \rho_t(x) - \phi_t(x) \partial_t \rho_t(x) \right] dx dt. \tag{57}$$

This is a convex objective in $\hat{\phi}$ whose minimizers satisfy

$$\nabla \cdot (\nabla \hat{\phi}_t \rho_t) = -\partial_t \rho_t. \tag{58}$$

This is (14) written in terms of $b_t(x) = \nabla \phi_t(x)$. The solution of this equation is unique up to a constant by the Fredholm alternative since its right hand-side satisfies the solvability condition $\int_{\mathbb{R}^d} \partial_t \rho_t(x) dx = 0$. □

**Derivation of (32).** If $\hat{b}_t(x) = \nabla \hat{\phi}_t(x)$, the SDEs (20) and (21) reduce to

$$dX_t^{\hat{b}} = -\varepsilon_t \nabla U_t(X_t^{\hat{b}}) dt + \hat{\nabla} \phi_t(X_t^{\hat{b}}) dt + \sqrt{2\varepsilon_t} dW_t, \qquad \hat{X}_0^{\hat{b}} \sim \rho_0, \tag{59}$$

$$dA_t^{\hat{b}} = \Delta \hat{\phi}_t(X_t^{\hat{b}}) dt - \nabla U_t(X_t^{\hat{b}}) \cdot \nabla \hat{\phi}(X_t^{\hat{b}}) dt - \partial_t U_t(X_t^{\hat{b}}) dt, \qquad A_0^{\hat{b}} = 0, \tag{60}$$

Since by Itô formula we have

$$d\hat{\phi}_t(X_t^{\hat{b}}) = \partial_t \hat{\phi}_t(X_t^{\hat{b}}) dt - \varepsilon_t \nabla \hat{\phi}_t(X_t^{\hat{b}}) \cdot \nabla U_t(X_t^{\hat{b}}) dt + |\nabla \hat{\phi}_t(X_t^{\hat{b}})|^2 dt$$
$$+ \sqrt{2\varepsilon_t} \nabla \hat{\phi}_t(X_t^{\hat{b}}) \cdot dW_t + \varepsilon_t \Delta \hat{\phi}_t(X_t^{\hat{b}}) dt, \tag{61}$$

we can express

$$\Delta \hat{\phi}_t(X_t^{\hat{b}}) dt = \frac{1}{\varepsilon_t} d\hat{\phi}_t(X_t^{\hat{b}}) dt - \frac{1}{\varepsilon_t} \partial_t \hat{\phi}_t(X_t^{\hat{b}}) dt + \nabla \hat{\phi}_t(X_t^{\hat{b}}) \cdot \nabla U_t(X_t^{\hat{b}}) dt$$
$$- \frac{1}{\varepsilon_t} |\nabla \hat{\phi}_t(X_t^{\hat{b}})|^2 dt - \sqrt{\frac{2}{\varepsilon_t}} \nabla \hat{\phi}_t(X_t^{\hat{b}}) \cdot dW_t. \tag{62}$$

If we insert this expression in the SDE (60), we can write it as

$$dA_t^{\hat{b}} = \frac{1}{\varepsilon_t} d\hat{\phi}_t(X_t^{\hat{b}}) dt + dB_t. \tag{63}$$

where $dB_t$ is given by (33). Integrating (63) gives (32).

## 4.2 TIME-DISCRETIZED VERSION OF PROPOSITION 3

Here we show how to generalize the result in Proposition 3 if we time discretize the SDE in (20) using Euler-Marayuma scheme and use some suitable time-discretized version of the ODE (21).

**Proposition 7.** *Let $0 = t_0 < t_1 < \cdots < t_K = 1$ be a time grid on $[0, 1]$, denote $\Delta t_k = t_{k+1} - t_k$ for $k = 0, \ldots, K - 1$, set $\tilde{X}_0^{\hat{b}} \sim \rho_0$ and $\tilde{A}_0^{\hat{b}} = 0$, and for $k = 0, \ldots, K - 1$ define $\tilde{X}_{t_{k+1}}^{\hat{b}}, \tilde{A}_{t_{k+1}}^{\hat{b}}$ recursively via*

$$\tilde{X}_{t_{k+1}}^{\hat{b}} = \tilde{X}_{t_k}^{\hat{b}} - \varepsilon_{t_k} \nabla U_{t_k}(\tilde{X}_{t_k}^{\hat{b}})\Delta t_k + \hat{b}_{t_k}(\tilde{X}_{t_k}^{\hat{b}})\Delta t_k + \sqrt{2\varepsilon_{t_k}}(W_{t_{k+1}} - W_{t_k}), \tag{64}$$

$$\tilde{A}_{t_{k+1}}^{\hat{b}} = \tilde{A}_{t_k}^{\hat{b}} + U_{t_k}(\hat{X}_{t_k}^{\hat{b}}) - U_{t_{k+1}}(\hat{X}_{t_{k+1}}^{\hat{b}}) + R_k^+(\tilde{X}_{t_k}^{\hat{b}}, \tilde{X}_{t_{k+1}}^{\hat{b}}) - R_k^-(\tilde{X}_{t_{k+1}}^{\hat{b}}, \tilde{X}_{t_k}^{\hat{b}}), \tag{65}$$

*where we defined*

$$R_k^{\pm}(x, y) = \frac{1}{4\varepsilon_{t_k}\Delta t_k}\big|y - x + \Delta t_k(\varepsilon_{t_k}\nabla U_{t_k}(x) \mp b_{t_k}(x))\big|^2 \tag{66}$$

*Then for all $k = 0, \ldots, K$ and any test function $h : \mathbb{R}^d \to \mathbb{R}$, we have*

$$\int_{\mathbb{R}^d} h(x)\rho_{t_k}(x)dx = \frac{\mathbb{E}[e^{\tilde{A}_{t_k}^{\hat{b}}}h(\tilde{X}_{t_k}^{\hat{b}})]}{\mathbb{E}[e^{\tilde{A}_{t_k}^{\hat{b}}}]}, \qquad Z_{t_k} = e^{-F_{t_k}} = \mathbb{E}[e^{\tilde{A}_{t_k}^{\hat{b}}}], \tag{67}$$

*where the expectations are taken over the law of $(\tilde{X}_{t_k}^{\hat{b}}, \tilde{A}_{t_k}^{\hat{b}})$*

Note that the weights in (67) correct for the bias coming for both the time evolution of $U_t(x)$ and the fact that the Euler-Maruyama update in (64) does not satisfy the detailed-balance condition locally. It cab be checked by direct calculation that (65) is a consistent time-discretization of the ODE (21).

*Proof.* For simplicity of notations we will prove (67) for $k = K$: the argument for all the other $k = 1, \ldots, K - 1$ is similar. The update rule in (65) implies that

$$\tilde{A}_{t_K}^{\hat{b}} = \sum_{k=0}^{K-1}\left(U_{t_k}(\hat{X}_{t_k}^{\hat{b}}) - U_{t_{k+1}}(\hat{X}_{t_{k+1}}^{\hat{b}}) + R_k^+(\tilde{X}_{t_k}^{\hat{b}}, \tilde{X}_{t_{k+1}}^{\hat{b}}) - R_k^-(\tilde{X}_{t_{k+1}}^{\hat{b}}, \tilde{X}_{t_k}^{\hat{b}})\right)$$

$$= U_0(\tilde{X}_{t_0}^{\hat{b}}) - U_K(\tilde{X}_{t_K}^{\hat{b}}) + \sum_{k=0}^{K-1}\left(R_k^+(\tilde{X}_{t_k}^{\hat{b}}, \tilde{X}_{t_{k+1}}^{\hat{b}}) - R_k^-(\tilde{X}_{t_{k+1}}^{\hat{b}}, \tilde{X}_{t_k}^{\hat{b}})\right), \tag{68}$$

Now, given the test function $h : \mathbb{R}^d \to \mathbb{R}$, consider

$$I[h] \equiv \mathbb{E}\big[e^{\tilde{A}_{t_K}^{\hat{b}}}h(\tilde{X}_{t_K}^{\hat{b}})\big] \tag{69}$$

Since the transition probability density function of the Euler-Maruyama update in (64) reads

$$\rho_{t_k}^+(x_{k+1}|x_k) = (4\pi\varepsilon_{t_k}\Delta t_k)^{-d/2}\exp\big(-R_k^+(x_k, x_{k+1})\big), \tag{70}$$

the joint probability density function of the path $(\tilde{X}_{t_0}^{\hat{b}}, \tilde{X}_{t_1}^{\hat{b}}, \ldots, \tilde{X}_{t_K}^{\hat{b}})$ is given by

$$\rho(x_0, \ldots, x_K) = \exp\left(-U_0(x_0) + F_0\right)\prod_{k=0}^{K-1}\rho_{t_k, \Delta t_k}^+(x_{k+1}|x_k)$$

$$= C\exp\left(-U_0(x_0) + F_0 - \sum_{k=0}^{K-1}R_k^+(x_k, x_{k+1})\right) \tag{71}$$

where $C = \prod_{k=0}^{K-1}(4\pi\varepsilon_{t_k}\Delta t_k)^{-d/2}$. We can use this density along with the explicit expression for $\tilde{A}_{T_K}^{\hat{b}}$ in (68) to express the expectation (69) as an integral over $\rho(x_0, x_1, \ldots, x_K)$

$$I[h] = C\int_{\mathbb{R}^{d(K+1)}}dx_0\cdots dx_K\exp\left(-U_0(x_0) + F_0 - \sum_{k=0}^{K-1}R_k^+(x_k, x_{k+1})\right)$$

$$\times \exp\left(U_0(x_0) - U_K(x_K) + \sum_{k=0}^{K-1}\left(R_k^+(x_k, x_{k+1}) - R_k^-(x_{k+1}, x_k)\right)\right)h(x_K) \tag{72}$$

where the second exponential comes from the factor $e^{\tilde{A}_K^{\hat{b}}}$. (72) simplifies into

$$I[h] = C \int_{\mathbb{R}^{d(K+1)}} dx_0 \cdots dx_K \exp\left(-U_K(x_K) + F_0 - \sum_{k=0}^{K-1} R_k^-(x_{k+1}, x_k)\right) h(x_K) \tag{73}$$

In this expression we recognize a product of factors involving

$$\rho_{t_k}^-(x_k|x_{k+1}) = (4\pi\varepsilon_{t_k})^{-d/2} \exp\left(-R_k^-(x_{k+1}, x_k)\right), \tag{74}$$

which is the transition probability density function of the time-reversed update

$$\tilde{Y}_{t_k}^{\hat{b}} = \tilde{Y}_{t_{k+1}}^{\hat{b}} - \varepsilon_{t_k}\nabla U_{t_k}(\tilde{Y}_{t_{k+1}}^{\hat{b}})\Delta t_k - \hat{b}_{t_k}(\tilde{Y}_{t_{k+1}}^{\hat{b}})\Delta t_k + \sqrt{2\varepsilon_{t_k}}(W_{t_{k+1}} - W_{t_k}). \tag{75}$$

This implies in particular that we can perform the integrals in (73) sequentially over $x_0$, $x_1$, .., $x_{K-1}$ to be left with

$$I[h] = \int_{\mathbb{R}^d} \exp\left(-U_K(x_K) + F_0\right) h(x_K) dx_K \tag{76}$$

Therefore

$$I[1] = \int_{\mathbb{R}^d} \exp\left(-U_K(x_K) + F_0\right) dx_K = e^{-F_K + F_0}, \tag{77}$$

which is the second equation in (67), and

$$\frac{I[h]}{I[1]} = e^{F_K - F_0} \int_{\mathbb{R}^d} \exp\left(-U_K(x_K) + F_0\right) h(x_K) dx_K = \int_{\mathbb{R}^d} h(x)\rho_{t_K}(x)dx \tag{78}$$

which is the first equation in (67). □

## 4.3 Solving for the optimal drift via Feynman-Kac formula

Without loss of generality, we can always look for a solution to (24) in the form of $b_t(x) = \nabla\phi_t(x)$, so that this equation becomes the Poisson equation

$$\Delta\phi_t - \nabla U_t \cdot \nabla\phi_t = \partial_t U_t - \partial_t F_t. \tag{79}$$

The solution to this equation can be expressed via Feynman-Kac formula:

**Proposition 8.** *Let $X_\tau^{t,x}$ satisfy the following SDE*

$$dX_\tau^{t,x} = -\nabla U_t(X_\tau^{t,x})d\tau + \sqrt{2}dW_\tau, \qquad X_{\tau=0}^{t,x} = x \tag{80}$$

*where $U_t$ is evaluated fixed at $t \in [0,1]$ fixed. Assume geometric ergodicity of the semi-group associated with (80), i.e. the probability distribution of the solutions to this SDE converges exponentially fast towards their unique equilibrium distribution with density $\rho_t(x)$. Then for all $(t,x) \in [0,1] \times \mathbb{R}^d$ we have*

$$\phi_t(x) = \int_0^\infty \mathbb{E}\left[\partial_t F_t - \partial_t U_t(X_\tau^{t,x})\right]d\tau \tag{81}$$

*where the expectation is taken over the law of $X_\tau^{t,x}$.*

*Proof.* By Ito formula,

$$\begin{aligned}
d\phi_t(X_\tau^{t,x}) &= \left(\Delta\phi_t(X_\tau^{t,x}) - \nabla U_t(X_\tau^{t,x}) \cdot \nabla\phi_t(X_\tau^{t,x})\right)d\tau + \sqrt{2}\nabla\phi_t(X_\tau^{t,x}) \cdot dW_\tau \\
&= \left(\partial_t U_t(X_\tau^{t,x}) - \partial_t F_t\right)d\tau + \sqrt{2}\nabla\phi_t(X_\tau^{t,x}) \cdot dW_\tau
\end{aligned} \tag{82}$$

where the differential is taken with respect to $\tau$ at $t$ fixed, and we used (79) to get the second equality. If we integrate this relation on $\tau \in [0,T]$ and take expectation, we deduce that

$$\mathbb{E}\left[\phi_t(X_T^{t,x})\right] - \phi_t(x) = \int_0^T \mathbb{E}\left[\partial_t U_t(X_\tau^{t,x}) - \partial_t F_t\right]d\tau \tag{83}$$

where we use Ito isometry to zero the expectation of the martingale term involving $\sqrt{2}\nabla\phi_t(X_\tau^{t,x}) \cdot dW_\tau$. If we let $T \to \infty$, by ergodicty the first term at the left hand side converges towards a constant independent of $(t,x)$ which we can neglect – this fixes the gauge of the solution to (79) which is unique only up to a constant. What remains in this limit is the expression (81). Note that the integral in this expression converges since $\mathbb{E}\left[\partial_t U_t(X_\tau^{t,x})\right] \to \partial_t F_t$ exponentially fast as $\tau \to \infty$ by assumption of geometric ergodicity. □

**Example: moving Gaussian distribution.**    Let us consider the case where

$$U_t(x) = \tfrac{1}{2}(x - b_t)^T A_t(x - b_t), \tag{84}$$

where $b_t \in \mathbb{R}^d$ is a time-dependent vector field and $A_t = A_t^T \in \mathbb{R}^d \times \mathbb{R}^d$ is a time-dependent positive-definite matrix: we assume that both $b_t$ and $A_t$ are $C^1$ in time, and also that $\dot{A}_t A_t = A_t \dot{A}_t$. The free energy in this example is

$$F_t = -\log Z_t, \qquad Z_t = (2\pi)^{d/2} |\det A_t|^{-1/2}, \tag{85}$$

so that

$$\partial_t U_t(x) = -\dot{b}_t^T A_t(x - b_t) + \tfrac{1}{2}(x - b_t)^T \dot{A}_t(x - b_t), \qquad \partial_t F_t = \tfrac{1}{2}\mathrm{tr}(A_t^{-1}\dot{A}_t). \tag{86}$$

In this case, the SDE (80) reads

$$dX_\tau^{t,x} = -A_t(X_\tau^{t,x} - b_t)d\tau + \sqrt{2}dW_\tau, \qquad X_{\tau=0}^{t,x} = x, \tag{87}$$

and its solution is

$$X_\tau^{t,x} = e^{-A_t\tau}x + \left(1 - e^{-A_t\tau}\right)b_t + \sqrt{2}\int_0^\tau e^{-A_t(\tau-\tau')}dW_{\tau'}. \tag{88}$$

This implies that (using Ito isometry)

$$\begin{aligned}
\mathbb{E}\left[\partial_t U_t(X_\tau^{t,x})\right] &= -\dot{b}_t^T A_t e^{-A_t\tau}(x - b_t) + \tfrac{1}{2}(x - b_t)^T e^{-A_t\tau}\dot{A}_t e^{-A_t\tau}(x - b_t) \\
&\quad + \int_0^\tau \mathrm{tr}\left(e^{-A_t\tau}\dot{A}_t e^{-A_t\tau}\right)d\tau \\
&= -\dot{b}_t^T A_t e^{-A_t\tau}(x - b_t) + \tfrac{1}{2}(x - b_t)^T e^{-A_t\tau}\dot{A}_t e^{-A_t\tau}(x - b_t) \\
&\quad + \tfrac{1}{2}\mathrm{tr}\left(A_t^{-1}\dot{A}_t\right) - \tfrac{1}{2}\mathrm{tr}\left(A_t^{-1}\dot{A}_t e^{-2A_t\tau}\right).
\end{aligned} \tag{89}$$

Therefore, from (81), we have (using also (85))

$$\begin{aligned}
\phi_t(x) &= \int_0^\infty \left(\dot{b}_t^T A_t e^{-A_t\tau}(x - b_t) - \tfrac{1}{2}(x - b_t)^T e^{-A_t\tau}\dot{A}_t e^{-A_t\tau}(x - b_t) \right. \\
&\qquad\qquad \left. + \tfrac{1}{2}\mathrm{tr}(A_t^{-1}\dot{A}_t e^{-2A_t\tau})\right)d\tau \\
&= \dot{b}_t \cdot (x - b_t) - \tfrac{1}{4}(x - b_t)^T \dot{A}_t A_t^{-1}(x - b_t) + \tfrac{1}{4}\mathrm{tr}(A_t^{-1}\dot{A}_t A_t^{-1}).
\end{aligned} \tag{90}$$

This solution checks out since it implies that

$$-\nabla U_t(x) \cdot \nabla \phi_t(x) + \Delta \phi_t(x) = -\dot{b}_t^T A_t(x - b_t) + \tfrac{1}{2}(x - b_t)^T \dot{A}_t(x - b_t) - \tfrac{1}{2}\mathrm{tr}(A_t^{-1}\dot{A}_t), \tag{91}$$

which is $\partial_t U_t(x) - \partial_t F_t$ as it should.

## 4.4    EXTENSIONS AND GENERALIZATIONS

### 4.4.1    INERTIAL NETS

It is straightforward to generalize Proposition 9 so that the stochastic dynamics involves some memory/inertia:

**Proposition 9.** *Let $(X_t^{\hat{b},\mu}, R_t^{\hat{b},\mu}, A_t^{\hat{b},\mu})$ solve the coupled system of SDE/ODE*

$$dX_t^{\hat{b},\mu} = \hat{b}_t(X_t^{\hat{b},\mu})dt + R_t^{\hat{b},\mu}dt, \qquad\qquad X_0^{\hat{b},\mu} \sim \rho_0, \tag{92}$$

$$dR_t^{\hat{b},\mu} = -\mu\nabla U_t(X_t^{\hat{b},\mu})dt - \mu\varepsilon_t^{-1}R_t^{\hat{b},\mu}dt + \mu\sqrt{2\varepsilon_t^{-1}}dW_t, \qquad R_0^{\hat{b},\mu} \sim N(0, \mu Id), \tag{93}$$

$$dA_t^{\hat{b},\mu} = \nabla \cdot \hat{b}_t(X_t^{\hat{b},\mu})dt - \nabla U_t(X_t^{\hat{b},\mu}) \cdot \hat{b}_t(X_t^{\hat{b},\mu})dt - \partial_t U_t(X_t^{\hat{b},\mu})dt, \quad A_0^{\hat{b},\mu} = 0, \tag{94}$$

*where $\varepsilon_t > 0$ is a time-dependent diffusion coefficient, $\mu \geq 0$ is a mobility coefficient, and $W_t \in \mathbb{R}^d$ is the Wiener process. Then for all $t \in [0, 1]$ and any test function $h : \mathbb{R}^d \to \mathbb{R}$, we have*

$$\int_{\mathbb{R}^d} h(x)\rho_t(x)dx = \frac{\mathbb{E}[e^{A_t^{\hat{b},\mu}}h(X_t^{\hat{b},\mu})]}{\mathbb{E}[e^{A_t^{\hat{b},\mu}}]}, \qquad Z_t/Z_0 = e^{-F_t+F_0} = \mathbb{E}[e^{A_t^{\hat{b},\mu}}], \tag{95}$$

*where the expectations are taken over the law of $(X_t^{\hat{b},\mu}, A_t^{\hat{b},\mu})$.*

The proof of this proposition can be found at the end of this subsection. Note that when $\hat{b} = b$, the solution to (24), (94) is simply

$$A_t^{b,\gamma} = -F_t + F_0, \tag{96}$$

i.e. the weights are again deterministic with zero variance. In general, $\hat{b}$ will not be the optimal one, in which case using the SDE in (92)-(94) gives us the extra parameter $\mu$ to play with post-training to improve the ESS. Below we show that (92)-(94) reduce to (20)-(21) in the limit as $\mu \to \infty$. It is also easy to see that, if we set $\mu = 0$ in (92)-(94), we simply get that $R_t^{\hat{b},\mu} = 0$ and hence (97) reduces to the ODE $dX_t^{\hat{b},\mu} = \hat{b}_t(X_t^{\hat{b},\mu})dt$. Finally it is worth noting that (92)-(93) can be cast into Langevin equations with some extra forces. Indeed, if we introduce the velocity $V_t^{\hat{b},\mu} = \hat{b}_t(X_t^{\hat{b},\mu}) + R_t^{\hat{b},\mu}$, (92)-(93) can be written as

$$dX_t^{\hat{b},\mu} = V_t^{\hat{b},\mu}dt \qquad\qquad X_0^{\hat{b},\mu} \sim \rho_0, \tag{97}$$

$$dV_t^{\hat{b},\mu} = -\mu\nabla U_t(X_t^{\hat{b},\mu})dt + \mu\varepsilon_t^{-1}\hat{b}_t(X_t^{\hat{b},\mu})dt - \partial_t\hat{b}_t(X_t^{\hat{b},\mu})dt$$
$$+ \nabla b_t(X_t^{\hat{b},\mu})V_t^{\hat{b},\mu}dt - \mu\varepsilon_t^{-1}V_t^{\hat{b},\mu}dt + \mu\sqrt{2\varepsilon_t^{-1}}dW_t, \quad V_0^{\hat{b},\mu} \sim N(\hat{b}_0(X_0^{\hat{b},\mu}), \mu\mathrm{Id}) \tag{98}$$

In these equations, the terms $\mu\varepsilon_t^{-1}\hat{b}_t - \partial_t\hat{b}_t$ can be interpreted as non-conservative forces added to $-\mu\nabla U_t$, and the term $\nabla b_t V_t^{\hat{b},\mu}$ as an extra friction term added to $-\mu\varepsilon_t^{-1}V_t^{\hat{b},\mu}$.

*Proof of Proposition 9.* Denote by $f_t^{\hat{b},\mu}(x, r, a)$ the joint PDF of $(X_t^{\hat{b},\mu}, R_t^{\hat{b},\mu}, A_t^{\hat{b},\mu})$. This PDF satisfies the FPE

$$\partial_t f_t^{\hat{b},\mu} = -\nabla_x \cdot ([\hat{b}_t + r]f_t^{\hat{b},\mu}) + \mu\nabla U_t \cdot \nabla_r f + \mu\varepsilon_t^{-1}\nabla_r \cdot (rf_t^{\hat{b},\mu} + \mu\nabla_r f_t^{\hat{b},\mu})$$
$$- (\nabla \cdot \hat{b}_t - \nabla U_t \cdot \hat{b}_t - \partial_t U_t)\partial_a f_t^{\hat{b},\mu}, \tag{99}$$

$$f_0^{\hat{b},\mu}(x, r, a) = \rho_0(x)(2\pi\mu)^{-d/2}e^{-|r|^2/(2\mu)}\delta(a).$$

Let

$$g_t^{\hat{b},\mu}(x, r) = \int_{\mathbb{R}} e^a f_t^{\hat{b},\mu}(x, r, a)da. \tag{100}$$

We can derive an equation for $g_t^{\hat{b},\mu}(x)$ by multiplying both sides of the FPE (99) by $e^a$ and integrating over $a \in \mathbb{R}$. Using equations similar to (44), we arrive at

$$\partial_t g_t^{\hat{b},\mu} = -\nabla_x \cdot ([\hat{b}_t + r]g_t^{\hat{b},\mu}) + \mu\nabla U_t \cdot \nabla_r f + \mu\varepsilon_t^{-1}\nabla_r \cdot (rg_t^{\hat{b},\mu} + \mu\nabla_r g_t^{\hat{b},\mu})$$
$$+ (\nabla \cdot \hat{b}_t - \nabla U_t \cdot \hat{b}_t - \partial_t U_t)g_t^{\hat{b},\gamma}, \tag{101}$$

$$g_0^{\hat{b},\mu}(x, r) = \rho_0(x)(2\pi\mu)^{-d/2}e^{-|r|^2/(2\mu)}.$$

Since $\rho_0(x) = e^{-U_0(x)+F_0}$, it can be checked by direct substitution that the solution to this equation is

$$g_t^{\hat{b},\mu}(x, r) = e^{-U_t(x)+F_0}(2\pi\mu)^{-d/2}e^{-|r|^2/(2\mu)}. \tag{102}$$

Therefore

$$\int_{\mathbb{R}^{2d}} g_t^{\hat{b},\mu}(x, r)dxdr = \int_{\mathbb{R}^{2d+1}} e^a f_t^{\hat{b},\mu}(x, r, a)dxdrda = e^{-F_t+F_0}, \tag{103}$$

where the first equality follows from the definition of $g_t^{\hat{b},\mu}$ and the second from its explicit expression and the definition of the free energy that implies $\int_{\mathbb{R}^d} e^{-U_t(x)}dx = e^{-F_t}$. Equation (103) is the second

equation in (95). From (102) we also deduce that, given any test function $h : \mathbb{R}^d \to \mathbb{R}$, we have

$$
\begin{aligned}
\frac{\int_{\mathbb{R}^{2d+1}} e^a h(x) f_t^{\hat{b},\mu}(x,r,a) dx dr da}{\int_{\mathbb{R}^{2d+1}} e^a f_t^{\hat{b},\mu}(x,r,a) dx dr da} &= \frac{\int_{\mathbb{R}^{2d}} h(x) g_t^{\hat{b},\mu}(x,r) dx dr}{\int_{\mathbb{R}^{2d}} g_t^{\hat{b},\mu}(x,r) dx dr} \\
&= \frac{\int_{\mathbb{R}^d} h(x) e^{-U_t(x)+F_0} dx}{\int_{\mathbb{R}^d} e^{-U_t(x)+F_0}} \\
&= \frac{\int_{\mathbb{R}^d} h(x) e^{-U_t(x)} dx}{\int_{\mathbb{R}^d} e^{-U_t(x)} dx} \\
&= e^{F_t} \int_{\mathbb{R}^d} h(x) e^{-U_t(x)} dx \\
&= \int_{\mathbb{R}^d} h(x) \rho_t(x) dx.
\end{aligned}
\tag{104}
$$

Since by definition of $f_t^{\hat{b},\mu}(x,r,a)$ the left hand-side of this equation can be expressed as the ratio of expectations over $(X_t^{\hat{b},\mu}, A_t^{\hat{b},\mu})$ in the first equation in (95) we are done. $\qquad\square$

To see what happens when $\mu \to \infty$, let us assume that $\varepsilon_t = \varepsilon$ (time-independent) and integrate (93) using Duhamel principle as

$$
R_t^{\hat{b},\mu} = e^{-\mu\varepsilon^{-1}t} R_0^{\hat{b},\mu} - \mu \int_0^t e^{-\mu\varepsilon^{-1}(t-s)} \nabla U_s(X_s^{\hat{b},\mu}) ds + \mu\sqrt{2\varepsilon^{-1}} \int_0^t e^{-\mu\varepsilon^{-1}(t-s)} dW_s, \tag{105}
$$

Letting $\mu \to \infty$, we see that the first term at the right hand side of (105) tends to zero, whereas the second one gives

$$
\lim_{\mu\to\infty} \mu \int_0^t e^{-\mu\varepsilon^{-1}(t-s)} \nabla U_s(X_s^{\hat{b},\mu}) ds = \varepsilon \nabla U_t(X_t^{\hat{b},\mu}) \tag{106}
$$

Finally, the third term at the right hand side of (105) is a Gaussian process with covariance

$$
C_{t,t'}^\mu = 2\mu^2\varepsilon^{-1} \int_0^{\min(t,t')} e^{-\mu\varepsilon^{-1}(t-s)-\mu\varepsilon^{-1}(t'-s)} ds = 2\mu \left( e^{-\mu\varepsilon^{-1}|t-t'|} - e^{-\mu\varepsilon^{-1}(t+t')} \right) \tag{107}
$$

As a result, given any test function $\phi_t$, we have

$$
\lim_{\mu\to\infty} \int_{[0,1]^2} \phi_t C_{t,t'}^\mu \phi_{t'} dt dt' = 2\varepsilon \int_0^1 \phi_t^2 dt \tag{108}
$$

which indicates that $C_{t,t'}^\mu$ converges weakly towards the Dirac distribution $\varepsilon\delta(t - t')$. Putting these results together shows that in the limit as $\mu \to \infty$, $R_t^{\hat{b},\mu} dt$ converges weakly towards $-\varepsilon \nabla U_t(X_t^{\hat{b},\mu}) dt + \sqrt{2\varepsilon} dW_t$, which, if inserted in (92), reduces this equation to (20). The case where $\varepsilon_t$ depends on time can be treated similarly.

### 4.4.2 MULTIMARGINAL NETS

Let $\mathcal{U}(\alpha, x)$ be a potential depending on $\alpha \in D \subset \mathbb{R}^N$ with $N \in \mathbb{N}$ as well as $x \in \mathbb{R}^d$, and assumed to be continuously differentiable in both arguments. Assume that $e^{-\mathcal{U}(\alpha,x)}$ is integrable in $x$ for all $\alpha \in D$, and define the family of PDF

$$
\varrho(\alpha, x) = e^{-\mathcal{U}(\alpha,x)+\mathcal{F}(\alpha)}, \qquad \mathcal{F}(\alpha) = -\log \int_{\mathbb{R}^d} e^{-\mathcal{U}(\alpha,x)} dx. \tag{109}
$$

Finally, define the family of matrix-valued $\hat{\mathcal{B}}(\alpha, x) : D \times \mathbb{R}^d \to \mathbb{R}^N \times \mathbb{R}^d$, assumed to be continuously differentiable in both arguments. These quantities allow us to give a generalization of Proposition 3 in which we can sample the PDF $\varrho(\alpha, x)$ along any differential path $\alpha_t \in D$:

**Proposition 10.** *Let $\alpha : [0,1] \to D$ be a differentiable path in $D$ and define the vector field $b : [0,1] \times \mathbb{R}^d \to \mathbb{R}^d$ as*

$$\hat{b}_t^\alpha(x) = \dot{\alpha}_t^T \hat{\mathcal{B}}(\alpha_t, x) \tag{110}$$

*as well as*

$$U_t^\alpha(x) = \mathcal{U}(\alpha_t, x), \qquad F_t^\alpha = \mathcal{F}(\alpha_t), \qquad \rho_t^\alpha = \varrho(\alpha, x) = e^{-U_t^\alpha(x) + F_t^\alpha} \tag{111}$$

*Let $(X_t^{\hat{b},\alpha}, A_t^{\hat{b},\alpha})$ solve the coupled system of SDE/ODE*

$$dX_t^{\hat{b},\alpha} = \hat{b}_t^\alpha(X_t^{\hat{b},\alpha})dt - \varepsilon_t \nabla U_t^\alpha(X_t^{\hat{b},\alpha})dt + \sqrt{2\varepsilon_t}dW_t \qquad X_0^{\hat{b},\alpha} \sim \rho_0^\alpha, \tag{112}$$

$$dA_t^{\hat{b},\alpha} = \nabla \cdot \hat{b}_t^\alpha(X_t^{\hat{b},\alpha})dt - \nabla U_t^\alpha(X_t^{\hat{b},\alpha}) \cdot \hat{b}_t^\alpha(X_t^{\hat{b},\alpha})dt - \partial_t U_t^\alpha(X_t^{\hat{b},\alpha})dt, \quad A_0^{\hat{b},\alpha} = 0, \tag{113}$$

*where $\varepsilon_t > 0$ is a time-dependent diffusion coefficient and $W_t \in \mathbb{R}^d$ is the Wiener process. Then for all $t \in [0,1]$ and any test function $h : \mathbb{R}^d \to \mathbb{R}$, we have*

$$\int_{\mathbb{R}^d} h(x)\rho_t^\alpha(x)dx = \frac{\mathbb{E}[e^{A_t^{\hat{b},\alpha}} h(X_t^{\hat{b},\alpha})]}{\mathbb{E}[e^{A_t^{\hat{b},\alpha}}]}, \qquad e^{-F_t^\alpha + F_0^\alpha} = \mathbb{E}[e^{A_t^{\hat{b},\alpha}}], \tag{114}$$

*where the expectations are taken over the law of $(X_t^{\hat{b},\alpha}, A_t^{\hat{b},\alpha})$.*

We will omit to give the proof of this proposition since it is a simple consequence of Proposition 3. The interest in formulating the problem in this new way is that is it easy to see that the right hand side of (113) (with $\partial_t F_t^\alpha$ added for convenience) can be written as

$$\nabla \cdot \hat{b}_t^\alpha(x) - \nabla U_t^\alpha(x) \cdot \hat{b}_t^\alpha(x) - \partial_t U_t^\alpha(x) + \partial_t F_t^\alpha$$
$$= \dot{\alpha}_t^T \left( \nabla_x \cdot \hat{\mathcal{B}}(\alpha_t, x) - \hat{\mathcal{B}}(\alpha_t, x)\nabla_x \mathcal{U}(\alpha_t, x) - \nabla_\alpha \mathcal{U}(\alpha_t, x) + \nabla_\alpha \mathcal{F}(\alpha_t) \right). \tag{115}$$

Therefore if we zero this term for all $(\alpha, x) \in D \times \mathbb{R}^d$ by picking the right $\hat{\mathcal{B}}(\alpha, x)$ we will obtain that (113) reduces to $dA_t^{\hat{b},\alpha} = -F_t^\alpha dt$, i.e. $A_t^{\hat{b},\alpha} = F_t^\alpha - F_0^\alpha$. Finding this optimal $\mathcal{B}(\alpha, x)$ can be obtained using the following result:

**Proposition 11** (Multimarginal PINN objective)**.** *Consider the objective for $(\hat{\mathcal{B}}, \hat{\mathcal{F}})$ given by:*

$$L_{PINN}^\alpha[\hat{\mathcal{B}}, \hat{\mathcal{F}}]$$
$$= \int_D \int_{\mathbb{R}^d} \left| \nabla_x \cdot \hat{\mathcal{B}}(\alpha, x) - \hat{\mathcal{B}}(\alpha, x)\nabla_x \mathcal{U}(\alpha, x) - \nabla_\alpha \mathcal{U}(\alpha, x) + \nabla_\alpha \hat{\mathcal{F}}(\alpha) \right|^2 \hat{\varrho}(\alpha, x)f(\alpha)dxd\alpha \tag{116}$$

*where $\hat{\varrho}(\alpha, x) > 0$ is a PDF in $x$ for all $\alpha \in D$, and $f(\alpha)$ is a PDF in $\alpha$. Then $\min_{\mathcal{B}, \mathcal{F}} L_{PINN}^\alpha[\hat{\mathcal{B}}, \hat{\mathcal{F}}] = 0$, and all minimizers $(\mathcal{B}, \mathcal{F})$ are such that and $\mathcal{B}(\alpha, x)$ solves*

$$\forall(\alpha, x) \in D \times \mathbb{R}^d : \quad 0 = \nabla_x \cdot \hat{\mathcal{B}}(\alpha, x) - \hat{\mathcal{B}}(\alpha, x)\nabla_x \mathcal{U}(\alpha, x) - \nabla_\alpha \mathcal{U}(\alpha, x) + \nabla_\alpha \mathcal{F}(\alpha), \tag{117}$$

*and $\mathcal{F}(\alpha)$ is the free energy (109) for all $\alpha \in D$.*

We will omit to give the proof of this proposition since it is a simple generalization of the proof of Proposition 4.

### 4.5 IMPLEMENTATION

The computation of the divergence $\nabla \cdot b_t(x)$ in the PINN objective given in (26) can be avoided by using Hutchinson's trace estimator, see Appendix 4.5.1. If we minimize (26) *off-policy*, i.e. with samples from some $\hat{\rho}_t \neq \rho_t$, this is perfectly valid, but may be inefficient for learning $\hat{b}_t$ over the support necessary for the problem. If we decide instead to set $\hat{\rho}_t(x) = \rho_t(x)$, since the SDEs in (20) and (21) can be used with any $\hat{b}_t(x)$ to estimate expectation over $\rho_t(x)$ via (22), we can write the PINN objective on-policy as

$$L_{PINN}^T[\hat{b}, \hat{F}] = \int_0^T \frac{1}{\mathbb{E}[e^{A_t^{\hat{b}}}]} \mathbb{E}\left[ e^{A_t^{\hat{b}}} \left| \nabla \cdot \hat{b}_t(X_t^{\hat{b}}) - \nabla U_t(X_t^{\hat{b}}) \cdot \hat{b}_t(X_t^{\hat{b}}) - \partial_t U_t(X_t^{\hat{b}}) + \partial_t \hat{F}_t \right|^2 \right] dt \tag{118}$$

These expectations can be estimated empirically over a population of solutions to (20) and (21). Crucially, since we can switch from off-policy to on-policy after taking the gradient of the PINN objective, *when computing the gradient of (118) over $\hat{b}_t(x)$, $(X_t^{\hat{b}}, A_t^{\hat{b}})$ can be considered independent of $\hat{b}_t(x)$ and do not need to be differentiated over.* In other words, the method does not require backpropagation through the simulation even if used on-policy, i.e. even though it uses the current value of $\hat{b}_t$ to estimate the loss and its gradient. Finally note that we can use the ODE (21) for $A_t^{\hat{b}}$ to write (118) as

$$L_{\text{PINN}}^T[\hat{b}, \hat{F}] = \int_0^T \frac{1}{\mathbb{E}[e^{A_t^{\hat{b}}}]} \mathbb{E}\left[e^{A_t^{\hat{b}}} |\partial_t A_t^{\hat{b}} + \partial_t \hat{F}_t|^2\right] dt \tag{119}$$

Since $\mathbb{E}[e^{A_t^{\hat{b}}} \partial_t A_t^{\hat{b}}]/\mathbb{E}[e^{A_t^{\hat{b}}}] = \partial_t \log \mathbb{E}[e^{A_t^{\hat{b}}}] = -\partial_t F_t$, (119) clearly shows that this loss controls the variance of $\partial_t A_t^{\hat{b}}$, which directly connects the Jarzynski weights to the PINN objective.

Learning $b_t(x)$ and $F_t$ for $t \in [0,1]$ from the start can be challenging if the initial $\hat{b}_t(x)$ is far from exact and the weights gets large variance as $t$ increases. This problem can be alleviated by estimating $b_t(x)$ sequentially. In practice, this amounts to annealing $T$ from a small initial value to $T = 1$, in such a way that $b_t(x)$ is learned sufficiently accurately so that variance of the weights remains small. This variance can be estimated on the fly, which also give us an estimate of the effective sample size (ESS) of the population at all times $t \in [0,1]$.

Note that we can also employ resampling strategies of the type used in SMC to keep the variance of the weights low Doucet et al. (2001); Bolić et al. (2004).

We can proceed similarly with the AM loss (31) by rewriting it as

$$L_{\text{AM}}^T[\hat{\phi}] = \int_0^T \frac{\mathbb{E}\left[e^{A_t^{\hat{b}}} \left[\frac{1}{2}|\nabla \hat{\phi}_t(X_t^{\hat{b}})|^2 + \partial_t \phi_t(X_t^{\hat{b}})\right]\right]}{\mathbb{E}[e^{A_t^{\hat{b}}}]} dt + \frac{\mathbb{E}\left[e^{A_0^{\hat{b}}} \phi_0(X_0^{\hat{b}})\right]}{\mathbb{E}[e^{A_0^{\hat{b}}}]} - \frac{\mathbb{E}\left[e^{A_T^{\hat{b}}} \phi_T(X_T^{\hat{b}})\right]}{\mathbb{E}[e^{A_T^{\hat{b}}}]}. \tag{120}$$

These expectations can be estimated empirically over solutions to (20) and (21) with $\hat{b}_t(x) = \nabla \hat{\phi}_t(x)$. The above implementation is detailed in Algorithm 1.

### 4.5.1 HUTCHINSON'S TRACE ESTIMATOR FOR THE EVALUATION OF $\nabla_t \hat{b}_t(x)$

It is well-known that, if $\nabla\nabla b_t(x)$ is bounded,

$$\nabla \cdot \hat{b}_t(x) = \frac{1}{2\delta}\mathbb{E}\left[\eta \cdot \left(\hat{b}_t(x + \delta\eta) - \hat{b}_t(x - \delta\eta)\right)\right] + O(\delta^2), \tag{121}$$

where $0 < \delta \ll 1$ is an adjustable parameter and $\eta \sim N(0, \text{Id})$. Indeed we have

$$\frac{1}{2\delta}\eta \cdot \left(\hat{b}_t(x + \delta\eta) - \hat{b}_t(x - \delta\eta)\right) = \eta^T \nabla b_t(x)\eta + O(\delta^2), \tag{122}$$

which implies (121) after taking the expectation over $\eta$.

We can use this formula to estimate the PINN loss via

$$L_{\text{PINN}}^{T,\delta}[\hat{b}, \hat{F}] = \int_0^T \mathbb{E}\left[R_t^\delta(x_t, \eta) R_t^\delta(x_t, \eta')\right] dt \tag{123}$$

where the expectation is now taken independent over $x_t \sim \hat{\rho}_t$, $\eta \sim N(0, \text{Id})$, and $\eta' \sim N(0, \text{Id})$, and we defined

$$R_t^\delta(x, \eta) = \frac{1}{2\delta}\eta \cdot \left(\hat{b}_t(x + \delta\eta) - \hat{b}_t(x - \delta\eta)\right) - \nabla U_t(x) \cdot \hat{b}_t(x) - \partial_t U_t(x) + \partial_t \hat{F}_t \tag{124}$$

The expectation in (123) is unbiased since $\eta \perp \eta'$, and its accuracy can be controlled by lowering $\delta$.

### 4.6 DETAILS ON NUMERICAL EXPERIMENTS

In the following we include details for reproducing the experiments presented in Section 3. An overview of the training procedure is given in Algorithm 1. Note that the SDE for the weights can replaced with (32) when learning with $\hat{\phi}_t$, as one would do with the action matching loss (31).

---

**Algorithm 1** Training: Note that for both objectives the resultant set of walkers across time slices $\{x_k^i\}$ are detached from the computational graph when taking a gradient step (*off-policy learning*).

---

1: **Initialize:** $n$ walkers, $x_0 \sim \rho_0$, $A_0 = 0$, $K$ time steps, model parameters for $\{\hat{b}_t, \hat{F}_t\}$ or $\hat{\phi}_t$ respectively, diffusion coefficient $\varepsilon_t$, learning rate $\eta$
2: **repeat**
3:     Randomize time grid: $t_0, t_1, \ldots, t_K \sim \text{Uniform}(0, T)$, sort such that $t_0 < t_1 < \cdots < t_K$
4:     **for** $k = 0, \ldots, K$ **do**
5:         $\Delta t_k = t_{k+1} - t_k,$
6:         **for** each walker $i = 1, \ldots, n$ **do**
7:             $x_{t_{k+1}}^i = x_{t_k}^i - \varepsilon_{t_k} \nabla U_{t_k}(x_{t_k}^i) \Delta t_k + \hat{b}_{t_k}(x_{t_k}^i) \Delta t_k + \sqrt{2\varepsilon_{t_k}} (W_{t_{k+1}}^i - W_{t_k}^i)$
8:             $A_{t_{k+1}}^i = A_{t_k}^i - \partial_t U_{t_k}(x_{t_k}^i) \Delta t_k - \hat{b}_{t_k}(x_{t_k}^i) \cdot \nabla U_{t_k}(x_{t_k}^i) \Delta t_k + \nabla \cdot \hat{b}_{t_k}(x_{t_k}^i) \Delta t_k$
9:         **end for**
10:     **end for**
11:     Estimate (118) or (120), respectively, by replacing the expectation by an empirical average over the $n$ walkers and the time integral by an empirical average over $t_0, \ldots, t_K$.
12:     Take gradient descent step to update the model parameters.
13: **until** converged

---

### 4.6.1 PERFORMANCE METRICS

**Effective sample size.** We can compute the self-normalized ESS as

$$\text{ESS}_t = \frac{\left(N^{-1} \sum_{i=1}^N \exp\left(A_t^i\right)\right)^2}{N^{-1} \sum_{i=1}^N \exp\left(2A_t^i\right)} \tag{125}$$

at time $t$ along the SDE trajectory. We can use the ESS both as a quality metric and as a trigger for when to perform resampling of the walkers based on the weights, using, e.g. systematic resampling (Doucet et al., 2001; Bolić et al., 2004). Systematic resampling is one of many resampling techniques from particle filtering wherein some walkers are killed and some are duplicated based on their importance weights.

**2-Wasserstein distance.** The 2-Wasserstein distance reported in Table 1 were computed with 2000 samples from the model and the target density using the Python Optimal Transport library.

**Maximum Mean Discrepancy (MMD).** We use the MMD code from Blessing et al. (2024) to benchmark the performance of NETS on Neal's funnel. We use the definition of the MMD as

$$\text{MMD}^2(\hat{\rho}, \rho) \approx \frac{1}{n(n-1)} \sum_{i,j}^n k(\hat{x}_i, \hat{x}_j) + \frac{1}{m(m-1)} \sum_{i,j}^m k(x_i, x_j) - \frac{2}{nm} \sum_i^n \sum_j^m k(\hat{x}_i, x_j) \tag{126}$$

where $\hat{x} \sim \hat{\rho}$ is from the model distribution and $x \sim \rho$ is from the target and $k : \mathbb{R}^d \times \mathbb{R}^d \to R$ is chosen to be the radial basis kernel with unit bandwidth.

### 4.6.2 40-MODE GMM

The 40-mode GMM is defined with the mean vectors given as:

$$\mu_1 = (-0.2995,\ 21.4577)\,, \qquad \mu_2 = (-32.9218,\ -29.4376)\,,$$
$$\mu_3 = (-15.4062,\ 10.7263)\,, \qquad \mu_4 = (-0.7925,\ 31.7156)\,,$$
$$\mu_5 = (-3.5498,\ 10.5845)\,, \qquad \mu_6 = (-12.0885,\ -7.8626)\,,$$
$$\mu_7 = (-38.2139,\ -26.4913)\,, \qquad \mu_8 = (-16.4889,\ 1.4817)\,,$$
$$\mu_9 = (15.8134,\ 24.0009)\,, \qquad \mu_{10} = (-27.1176,\ -17.4185)\,,$$
$$\mu_{11} = (14.5287,\ 33.2155)\,, \qquad \mu_{12} = (-8.2320,\ 29.9325)\,,$$
$$\mu_{13} = (-6.4473,\ 4.2326)\,, \qquad \mu_{14} = (36.2190,\ -37.1068)\,,$$
$$\mu_{15} = (-25.1815,\ -10.1266)\,, \qquad \mu_{16} = (-15.5920,\ 34.5600)\,,$$
$$\mu_{17} = (-25.9272,\ -18.4133)\,, \qquad \mu_{18} = (-27.9456,\ -37.4624)\,,$$
$$\mu_{19} = (-23.3496,\ 34.3839)\,, \qquad \mu_{20} = (17.8487,\ 19.3869)\,,$$
$$\mu_{21} = (2.1037,\ -20.5073)\,, \qquad \mu_{22} = (6.7674,\ -37.3478)\,,$$
$$\mu_{23} = (-28.9026,\ -20.6212)\,, \qquad \mu_{24} = (25.2375,\ 23.4529)\,,$$
$$\mu_{25} = (-17.7398,\ -1.4433)\,, \qquad \mu_{26} = (25.5824,\ 39.7653)\,,$$
$$\mu_{27} = (15.8753,\ 5.4037)\,, \qquad \mu_{28} = (26.8195,\ -23.5521)\,,$$
$$\mu_{29} = (7.4538,\ -31.0122)\,, \qquad \mu_{30} = (-27.7234,\ -20.6633)\,,$$
$$\mu_{31} = (18.0989,\ 16.0864)\,, \qquad \mu_{32} = (-23.6941,\ 12.0843)\,,$$
$$\mu_{33} = (21.9589,\ -5.0487)\,, \qquad \mu_{34} = (1.5273,\ 9.2682)\,,$$
$$\mu_{35} = (24.8151,\ 38.4078)\,, \qquad \mu_{36} = (-30.8249,\ -14.6588)\,,$$
$$\mu_{37} = (15.7204,\ 33.1420)\,, \qquad \mu_{38} = (34.8083,\ 35.2943)\,,$$
$$\mu_{39} = (7.9606,\ -34.7833)\,, \qquad \mu_{40} = (3.6797,\ -25.0242)$$

These means follow the definition given in the FAB (Midgley et al., 2023) code base that has been subsequently used in recent papers. The time-dependent potential $U_t(x)$ is given by the interpolation of means.

### 4.6.3 NEAL'S 10-$d$ FUNNEL

The Neal's Funnel distribution is a 10-$d$ probability distribution defined as

$$x_0 \sim \mathsf{N}(0, \sigma^2), \quad x_{1:9} \sim \mathsf{N}(0, e^{x_0}) \tag{127}$$

where $\sigma = 3$ and we use subscripts here as a dimensional index and not as a time index like in the rest of the paper. Following this, we use as a definition of the interpolating potential:

$$U_t(x) = \frac{1}{2}x_0^2(1 - t + \frac{t}{\sigma^2}) + \frac{1}{2}\sum_{i=1}^{d-1} e^{-tx_0} x_i^2 + (d-1)tx_0 \tag{128}$$

so that at $t = 0$, we have $U_0(x) = \frac{1}{2}x_0^2 + \frac{1}{2}\sum_{i=1}^{d-1} x_i^2$ and at time $t = 1$ we have the funnel potential given as $U_1(x) = \frac{1}{2\sigma^2}x_0 + \frac{1}{2}\sum_{i=1}^{d-1} e^{-x_0} x_i^2 + (d-1)x_0$.

### 4.6.4 50-$d$ MIXTURE OF STUDENT-T DISTRIBUTIONS

Following Blessing et al. (2024), we use their mixture of 10 student-T distributions in 50 dimensions. We construct $U_t$ via interpolation of means from a single standard student-T distribution (mean 0). We use the same neural network as used in the GMM experiments.

To further drive home the fact that our annealed Langevin dynamics with transport can be taken post-training to the $\epsilon \to \infty$ limit to approach perfect sampling, we provide the following ablation from our model learned with the action matching loss given in Figure 4.

### 4.7 DETAILS OF THE $\varphi^4$ MODEL

We consider the Euclidean scalar $\phi^4$ theory given by the action

$$S_{\text{Euc}}[\varphi] = \int \left[ \partial_\mu \varphi(x)\partial^\mu \varphi(x) + m^2\varphi^2(x) + \lambda\varphi^4(x) \right] d^D x \tag{129}$$

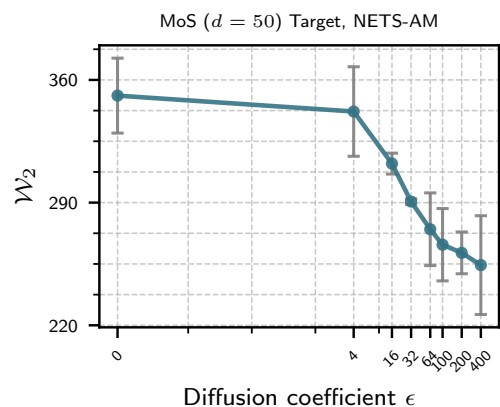

Figure 4: Reduction in $\mathcal{W}_2$ distance from taking the $\epsilon \to \infty$ limit in sampling with NETS. Note that the resolution of the SDE integration must increase to accommodate the higher stochasticity. Average taken over 3 sampling runs of 2000 walkers each.

where we use Einstein summation to denote the dot product with respect to the Euclidean metric and $D$ is the spacetime dimension. We are interested in acquiring a variant of this expression that provides a fast computational realization when put onto the lattice. Using Green's identity (integrating by parts) we note that

$$\int (\partial_\mu \varphi(x) \partial^\mu \varphi(x) d^d x = \int \partial_\mu \varphi \cdot \partial_\mu \varphi \, d^d x = -\int \varphi(x) \partial_\mu \partial^\mu \varphi(x) d^d x \ + \text{vanishing surface term}$$
(130)

so that

$$S_{\text{Euc}}[\varphi] = \int -\varphi(x)\partial_\mu \partial^\mu \varphi(x) + m^2 \varphi^2(x) + \lambda \varphi^4(x) \, d^d x.$$
(131)

Discretizing $S_{\text{Euc}}$ onto the lattice

$$\Lambda = \{a(n_0, \ldots, n_{d-1}) \mid n_i \in \{0, 1, 2, \ldots, L\}, \ i = 0, 1, \ldots, d, a \in \mathbb{R}_+\},$$

where $a$ is the lattice spacing used to define the physical point $x = an$, we use the forward difference operator to define

$$\partial_\mu \varphi(x) \to \tfrac{1}{a}[\varphi(x + \mu) - \varphi(x)] \qquad \partial_\mu \partial^\mu \varphi(x) \to \tfrac{1}{a^2}[\varphi(x + \mu) - 2\varphi(x) + \varphi(x - \mu)].$$
(132)

Using these expressions, we write the discretized lattice action as

$$S_{\text{Lat}} = \sum_{x \in \Lambda} a^D \left[ \sum_{\mu=1}^{D} -\tfrac{1}{a^2}[\varphi_{x+\mu}\varphi_x - 2\varphi_x^2 + \varphi_{x-\mu}\varphi_x] + m^2 \varphi_x^2 + \lambda \varphi_x^4 \right]$$
(133)

$$= \sum_x a^D \left[ 2Da^{-2}\varphi_x^2 - a^{-2} \sum_\mu [\varphi_{x+\mu}\varphi_x + \varphi_{x-\mu}\varphi_x] + m^2 \varphi_x^2 + \lambda \varphi_x^4 \right]$$
(134)

$$= \sum_x a^d \left[ 2Da^{-2}\varphi_x^2 - 2a^{-2} \sum_\mu [\varphi_x\varphi_{x+\mu}] + m^2 \varphi_x^2 + \lambda \varphi_x^4 \right]$$
(135)

$$= \sum_x a^D \left[ -2a^{-2} \sum_\mu \varphi_x\varphi_{x+\mu} + (2a^{-2}D + m^2)\varphi_x^2 + \lambda \varphi_x^4 \right]$$
(136)

where we have used the fact that on the lattice $\sum_x \varphi_x \varphi_{x+\hat{\mu}} = \sum_x \varphi_{x-\hat{\mu}}\varphi_x$ to get the third equality. It is useful to put the action in a form that is independent of the lattice spacing $a$. To do so, we introduce the re-scaled lattice field as

$$\varphi_x \to a^{D/2-1}\varphi_x, \quad m^2 \to a^2 m^2, \quad \text{and} \quad \lambda \to a^{4-D}\lambda.$$
(137)

Plugging these rescalings into (136) gives us the final expression

$$S_{\text{Lat}} = \sum_x \left[ -2 \sum_\mu \varphi_x \varphi_{x+\mu} \right] + (2D + m^2)\varphi_x^2 + \lambda \varphi_x^4, \tag{138}$$

which we are to use in simulation.

### 4.7.1 Free theory $\lambda = 0$

Turning off the interaction makes it possible to analytically solve the theory. To do this, introduce the discrete Fourier transform relations

$$\varphi_k = \frac{1}{\sqrt{L^D}} \sum_x \varphi_x e^{-ik \cdot x} \tag{139}$$

$$\varphi_x = \frac{1}{\sqrt{L^D}} \sum_k \varphi_k e^{ik \cdot x} \tag{140}$$

for discrete wavenumbers $k = \frac{2l\pi}{L}$ with $l = 0, \cdots, L-1$. Plugging in (140) into the first part of (136), we get the expanded sum

$$\sum_x \left[ -2 \sum_\mu \hat{\varphi}_x \hat{\varphi}_{x+\mu} \right] \rightarrow -\frac{2}{L^d} \sum_x \sum_\mu \sum_k \sum_{k'} \varphi_k \varphi_{k'} e^{i(k+k') \cdot x} e^{ik' \cdot \mu} \tag{141}$$

$$= -2 \sum_\mu \sum_k \sum_{k'} \delta_{k,-k'} \varphi_k \varphi_{k'} e^{ik' \cdot \mu} \tag{142}$$

$$= -2 \sum_\mu \sum_k \varphi_k \varphi_{-k} e^{-ik_\mu} \tag{143}$$

$$= -2 \sum_\mu \sum_k \varphi_k \varphi_{k^*} e^{-ik_\mu} \tag{144}$$

$$= -2 \sum_\mu \sum_k |\varphi_k|^2 [\cos k_\mu + i \sin k_\mu] = -\sum_\mu \sum_k |\varphi_k|^2 \cos k_\mu \tag{145}$$

where $\phi^*$ indicates conjugation, and we got the first equality by the orthogonality of the Fourier modes, the second by the Kronecker delta, and the third by the reality of the scalar field. Proceeding similarly for the terms proportional to $\varphi^2$ gives us the expression

$$S_k = \sum_k \left[ m^2 + 2D - 2 \sum_\mu \cos k_\mu \right] |\varphi|^2 \tag{146}$$

The above equation can be written in quadratic form to highlight that the field may be sampled analytically

$$S_k = \frac{1}{L^d} \sum_k \varphi_k M_{k,-k} \varphi_{-k} \tag{147}$$

$$\text{where} \quad M_{k,-k} = \left[ m^2 + 2D - 2 \sum_\mu \cos k_\mu \right] \delta_{k,-k} \tag{148}$$

Note that this free theory can be sampled for any $m^2 > 0$.

### 4.7.2 $\varphi^4$ numerical details

We numerically realize the above lattice theory in D=2 spacetime dimensions. We use an interpolating potential with time dependent $m_t^2 = (1-t)m_0^2 + tm_1^2$, $\lambda_t = (1-t)\lambda_0 + t\lambda_1$ where $\lambda_0$ is always chosen to be 0 (though we note that you could run this sampler for any $U_0$ that you could sample from easily, not just analytically but also with existing MCMC methods). For the $L = 20$ (d = $L \times L = 400$ dimensional) experiments, we identify the critical point of the theory (where the lattices go from ordered to disordered) using HMC by studying the distribution of the magnetization of the

field configurations as $M[\varphi^i(x)] = \sum_x \varphi^i(x)$, where summation is taken over all lattice sites on the $i^{th}$ lattice configuration. We identify this at $m_1^2 = -1.0$, $\lambda_1 = 0.9$ and use these as the target theory parameters on which to perform the sampling. For the $L = 16$ test ($d = 256$), we go past this phase transition into the ordered phase of the theory, which we identify via HMC simulations at $m_1^2 = -1.0$, $\lambda_1 = 0.8$.

## 4.8 LINK WITH VARGAS ET AL. (2024)

Consider the process $Y_t^{\hat{b}}$ solution to the SDE

$$dY_t^{\hat{b}} = \varepsilon_t \nabla U_t(Y_t^{\hat{b}})dt + 2\varepsilon_t \nabla \log \rho_t^{\hat{b}}(Y_t^{\hat{b}})dt + \hat{b}_t(Y_t^{\hat{b}})dt + \sqrt{2\varepsilon_t}dW_t, \quad Y_0^{\hat{b}} \sim \rho_0. \tag{149}$$

where $\rho_t^{\hat{b}}$ denotes the PDF of the process $X_t^{\hat{b}}$ defined by the SDE (20), i.e. the solution to the FPE

$$\partial_t \rho_t^{\hat{b}} = \varepsilon_t \nabla \cdot (\nabla U_t \rho_t^{\hat{b}} + \nabla \rho_t^{\hat{b}}) - \nabla \cdot (\hat{b}_t \rho_t^{\hat{b}}), \qquad \rho_0^{\hat{b}} = \rho_0 \tag{150}$$

The process $Y_t^{\hat{b}}$ has a simple interpretation: it is the time-reversed of the process run using the time-reversed potential $U_{1-t}$ and $-\hat{b}_{1-t}$: that is, if the additional drift $\hat{b}_t$ was the perfect one solution to (24), the law of $X^{\hat{b}} = (X_t^{\hat{b}})_{t \in [0,1]}$ and $Y^b = (Y_t^{\hat{b}})_{t \in [0,1]}$ should coincide. This suggests to learn $b$ using as objective a divergence of the path measure of $X^{\hat{b}}$ from that of $Y^{\hat{b}}$. This is essentially what is suggested in Vargas et al. (2024), and for the reader convenience let us re-derive some of their results in our notations.

The Kullback-Leibler divergence (or relative entropy) of the path measure of $X^{\hat{b}}$ from that of $Y^{\hat{b}}$ reads

$$\mathrm{KL}(X^{\hat{b}} \| Y^{\hat{b}}) = \frac{1}{4}\varepsilon_t \int_0^1 \mathbb{E}\big[|\nabla U_t(X_t^{\hat{b}}) + \nabla \log \rho_t^{\hat{b}}(X_t^{\hat{b}})|^2\big]dt \tag{151}$$

This objective is akin to the one used in score-based diffusion modeling (SBDM) and simply says that one way to adjust $\hat{b}$ is by matching the score of $\rho_t^{\hat{b}}$ to that of $\rho_t$. As written (151) is not explicit since we do not know $\nabla \log \rho_t^{\hat{b}}$. We can however make it explicit after a few manipulations similar to those used in SBDM. To this end, notice first that, by Ito formula, we have

$$d \log \rho_t^{\hat{b}}(X_t^{\hat{b}}) = \nabla \log \rho_t^{\hat{b}}(X_t^{\hat{b}}) \cdot (-\varepsilon_t \nabla U_t(X_t^{\hat{b}}) + \hat{b}_t(X_t^{\hat{b}}))dt + \varepsilon_t \Delta \log \rho_t^{\hat{b}}(X_t^{\hat{b}})dt \\ + \sqrt{2\varepsilon_t} \nabla \log \rho_t^{\hat{b}}(X_t^{\hat{b}}) \cdot dW_t \tag{152}$$

which implies that

$$\varepsilon_t \nabla \log \rho_t^{\hat{b}}(X_t^{\hat{b}}) \cdot \nabla U_t((X_t^{\hat{b}})dt = -d \log \rho_t^{\hat{b}}(X_t^{\hat{b}}) + \nabla \log \rho_t^{\hat{b}}(X_t^{\hat{b}}) \cdot \hat{b}_t(X_t^{\hat{b}})dt \\ + \varepsilon_t \Delta \log \rho_t^{\hat{b}}(X_t^{\hat{b}})dt + \sqrt{2\varepsilon_t} \nabla \log \rho_t^{\hat{b}}(X_t^{\hat{b}}) \cdot dW_t \tag{153}$$

Inserting this expression in (151) after expanding the square, and noticing that the martingale term involving $dW_t$ disappears by Ito isometry and that the term $d \log \rho_t^{\hat{b}}(X_t^{\hat{b}})$ can be integrated in time we arrive at

$$\mathrm{KL}(X^{\hat{b}} \| Y^{\hat{b}}) = \frac{1}{4}\varepsilon_t \int_0^1 \mathbb{E}\big[|\nabla U_t(X_t^{\hat{b}})|^2 + |\nabla \log \rho_t^{\hat{b}}(X_t^{\hat{b}})|^2 + 2\nabla U_t(X_t^{\hat{b}}) \cdot \nabla \log \rho_t^{\hat{b}}(X_t^{\hat{b}})\big]dt$$

$$= \frac{1}{4} \int_0^1 \mathbb{E}\big[\varepsilon_t|\nabla U_t(X_t^{\hat{b}})|^2 + \varepsilon_t|\nabla \log \rho_t^{\hat{b}}(X_t^{\hat{b}})|^2 + \varepsilon_t \nabla U_t(X_t^{\hat{b}}) \cdot \nabla \log \rho_t^{\hat{b}}(X_t^{\hat{b}})\big]dt$$

$$+ \frac{1}{4} \int_0^1 \mathbb{E}\big[\nabla \log \rho_t^{\hat{b}}(X_t^{\hat{b}}) \cdot \hat{b}_t(X_t^{\hat{b}}) + \varepsilon_t \Delta \log \rho_t^{\hat{b}}(X_t^{\hat{b}})\big]dt$$

$$+ \frac{1}{4}\mathbb{E}[\log \rho_0(X_0)] - \frac{1}{4}\mathbb{E}[\log \rho_1^{\hat{b}}(X_1)] \tag{154}$$

where we used $\rho_0^{\hat{b}} = \rho_0$. We can now use the following identities, each obtained using $\rho_t^{\hat{b}} \nabla \log \rho_t^{\hat{b}} = \nabla \rho_t^{\hat{b}}$ and one integration by parts:

$$
\begin{aligned}
\mathbb{E}\big[|\nabla \log \rho_t^{\hat{b}}(X_t^{\hat{b}})|^2\big] &= \int_{\mathbb{R}^d} |\nabla \log \rho_t^{\hat{b}}(x)|^2 \rho_t^{\hat{b}}(x)dx \\
&= \int_{\mathbb{R}^d} \nabla \log \rho_t^{\hat{b}}(x) \cdot \nabla \rho_t^{\hat{b}}(x)dx \\
&= -\int_{\mathbb{R}^d} \Delta \log \rho_t^{\hat{b}}(x) \rho_t^{\hat{b}}(x)dx \\
&= -\mathbb{E}\big[\Delta \log \rho_t^{\hat{b}}(X_t^{\hat{b}})\big],
\end{aligned}
\tag{155}
$$

$$
\begin{aligned}
\mathbb{E}\big[\nabla U_t(X_t^{\hat{b}}) \cdot \nabla \log \rho_t^{\hat{b}}(X_t^{\hat{b}})\big] &= \int_{\mathbb{R}^d} \nabla U_t(x) \cdot \nabla \log \rho_t^{\hat{b}}(x) \rho_t^{\hat{b}}(x)dx \\
&= \int_{\mathbb{R}^d} \nabla U_t(x) \cdot \nabla \rho_t^{\hat{b}}(x)dx \\
&= -\int_{\mathbb{R}^d} \Delta U_t(x) \rho_t^{\hat{b}}(x)dx \\
&= -\mathbb{E}\big[\Delta U_t(X_t^{\hat{b}})\big]
\end{aligned}
\tag{156}
$$

and

$$
\begin{aligned}
\mathbb{E}\big[\nabla \log \rho_t^{\hat{b}}(X_t^{\hat{b}}) \cdot \hat{b}_t(X_t^{\hat{b}})\big] &= \int_{\mathbb{R}^d} \nabla \log \rho_t^{\hat{b}}(x) \cdot \hat{b}_t(x) \rho_t^{\hat{b}}(x)dx \\
&= -\int_{\mathbb{R}^d} \nabla \cdot \hat{b}_t(x) \rho_t^{\hat{b}}(x)dx \\
&= -\mathbb{E}\big[\nabla \cdot \hat{b}_t(X_t^{\hat{b}})\big]
\end{aligned}
\tag{157}
$$

Inserting these expressions in (154), it reduces to

$$
\begin{aligned}
\mathrm{KL}(X^{\hat{b}}\|Y^{\hat{b}}) &= \frac{1}{4} \int_0^1 \mathbb{E}\big[\varepsilon_t |\nabla U_t(X_t^{\hat{b}})|^2 - \varepsilon_t \Delta U_t(X_t^{\hat{b}}) - \nabla \cdot b_t(X_t^{\hat{b}})\big]dt \\
&\quad + \frac{1}{4}\mathbb{E}[\log \rho_0(X_0)] - \frac{1}{4}\mathbb{E}[\log \rho_1^{\hat{b}}(X_1)]
\end{aligned}
\tag{158}
$$

This objective is still not practical because it involves $\log \rho_1^{\hat{b}}$, which is unknown. There is however a simple way to fix this, by adding a term in the Kullback-Leibler divergence (151)

$$
\mathrm{KL}'(X^{\hat{b}}\|Y^{\hat{b}}) = \mathrm{KL}(X^{\hat{b}}\|Y^{\hat{b}}) + \frac{1}{4}\mathbb{E}[\log(\rho_1^{\hat{b}}(X_1^{\hat{b}})/\rho_1(X_1^{\hat{b}}))]
\tag{159}
$$

This additional term is proportional to the Kullback-Leibler divergence of $\rho_1^{\hat{b}}$ from the target PDF $\rho_1$. Using (158) as well as $\rho_1(x) = e^{-U_1(x)+F_1}$, we can now express (159) as

$$
\begin{aligned}
\mathrm{KL}'(X^{\hat{b}}\|Y^{\hat{b}}) &= \frac{1}{4} \int_0^1 \mathbb{E}\big[\varepsilon_t |\nabla U_t(X_t^{\hat{b}})|^2 - \varepsilon_t \Delta U_t(X_t^{\hat{b}}) - \nabla \cdot b_t(X_t^{\hat{b}})\big]dt \\
&\quad + \frac{1}{4}\mathbb{E}[\log \rho_0(X_0)] + \frac{1}{4}\mathbb{E}[U_1(X_1)] - \frac{1}{4}F_1.
\end{aligned}
\tag{160}
$$

This is Equation (24) in Vargas et al. (2024) in which we set $\nabla \hat{\phi}_t(x) = \hat{b}_t(x)$ and we used that, for any $c_t : \mathbb{R}^d \to \mathbb{R}^d$, we have

$$
\mathbb{E} \int_0^1 c_t(X_t^{\hat{b}}) \cdot \overleftarrow{d}W_t = \sqrt{2\varepsilon_t} \int_0^1 \mathbb{E}\big[\nabla \cdot c_t(X_t^{\hat{b}})\big]dt.
\tag{161}
$$

Note that we can neglect the term $\frac{1}{4}\mathbb{E}[\log \rho_0(X_0)]$ in (158) since it does not depend on $\hat{b}$, so that the minimization of (158) can be cast into the minimization of (after multiplication by 4)

$$
\begin{aligned}
&\int_0^1 \mathbb{E}\big[\varepsilon_t |\nabla U_t(X_t^{\hat{b}})|^2 - \varepsilon_t \Delta U_t(X_t^{\hat{b}}) - \nabla \cdot b_t(X_t^{\hat{b}})\big] + \mathbb{E}[U_1(X_1) - F_1] \\
&= \int_0^1 \int_{\mathbb{R}^d} \big[\varepsilon_t |\nabla U_t(x)|^2 - \varepsilon_t \Delta U_t(x) - \nabla \cdot b_t(x))\big] \rho_t^{\hat{b}}(x)dx + \int_{\mathbb{R}^d} (U_1(x) - F_1)\rho_1^{\hat{b}}(x)dx
\end{aligned}
\tag{162}
$$

where $\rho_t^{\hat{b}}$ solves (150).

Let us check that the minimizer of (162) is $\hat{b}_t = b_t$, the solution to (24), so that we also have $\rho_t^{\hat{b}} = \rho_t^b = \rho_t$. To this end, notice that the minimization of (162) can be performed with the method Lagrange multiplier, using the extended objective

$$
\int_0^1 \int_{\mathbb{R}^d} \left[ \varepsilon_t |\nabla U_t(x)|^2 - \varepsilon_t \Delta U_t(x) - \nabla \cdot \hat{b}_t(x) \right] \rho_t^{\hat{b}}(x) dx dt + \int_{\mathbb{R}^d} (U_1(x) - F_1) \rho_1^{\hat{b}}(x) dx
$$

$$
+ \int_0^1 \int_{\mathbb{R}^d} \lambda_t(x) \left( \partial_t \rho_t^{\hat{b}} - \varepsilon_t \nabla \cdot (\nabla U_t \rho_t^{\hat{b}} + \nabla \rho_t^{\hat{b}}) + \nabla \cdot (\hat{b}_t \rho_t^{\hat{b}}) \right) dx dt
$$

(163)

where $\lambda_t(x)$ is the Lagrange multiplier to be determined. Taking the first variation of this objective over $\lambda$, $\rho^{\hat{b}}$, and $\hat{b}$, we arrive at the Euler-Lagrange equations

$$
\begin{aligned}
0 &= \partial_t \rho_t^{\hat{b}} - \varepsilon_t \nabla \cdot (\nabla U_t \rho_t^{\hat{b}} + \nabla \rho_t^{\hat{b}}) + \nabla \cdot (\hat{b}_t \rho_t^{\hat{b}}), & \rho_0^{\hat{b}} &= \rho_0, \\
0 &= \varepsilon_t |\nabla U_t|^2 - \varepsilon_t \Delta U_t - \nabla \cdot \hat{b}_t \\
&\quad - \partial_t \lambda_t + \varepsilon_t \nabla U_t \cdot \nabla \lambda_t - \varepsilon_t \Delta \lambda_t - \hat{b}_t \cdot \nabla \lambda_t & \lambda_1 &= -U_1 + F_1 \\
0 &= \nabla \rho_t^{\hat{b}} - \rho_t^{\hat{b}} \nabla \lambda_t
\end{aligned}
$$

(164)

We can check that $\hat{b}_t(x) = b_t(x)$, $\rho_t^{\hat{b}}(x) = \rho_t(x) = e^{-U_t(x) + F_t}$, and $\lambda_t(x) = -U_t(x) + F_t$ is a solution: indeed this solves the first and the last equations in (164) and reduces the second to

$$
\begin{aligned}
0 &= \left[ \varepsilon_t |\nabla U_t|^2 - \varepsilon_t \Delta U_t - \nabla \cdot b_t \right] \\
&\quad + \partial_t U_t - \partial_t F_t - \varepsilon_t |\nabla U_t|^2 + \Delta U_t + \nabla b_t \cdot \nabla U_t \\
&= -\nabla \cdot b_t + \partial_t U_t - \partial_t F_t + \nabla b_t \cdot \nabla U_t
\end{aligned}
$$

(165)

which is satisfied since $b_t$ solves (24).

