# OpenReview forum: "NETS: A Non-Equilibrium Transport Sampler"
_ICLR.cc/2025/Conference — Submitted to ICLR 2025_

### Official Review · Reviewer_QP9k · 2024-10-28

**Soundness:** 3
**Presentation:** 3
**Contribution:** 4
**Rating:** 8
**Confidence:** 2

**Summary:**

The paper proposes a novel sampler for unnormalized probability distributions by augmenting the Stochastic Differential Equations (SDE) framework of Annealed Importance Sampling (AIS). The method introduces an additional drift term in the SDE, which enhances the effective sample size of the collected samples, thus reducing bias. The drift term can be estimated through either a Physics-Informed Neural Network (PINN) or Action matching objectives, which encourages the SDE to be nearly unbiased.

**Strengths:**

The paper presents a diffusion-based, simulation-free method for multi-modal sampling, which demonstrates significant potential in sampling efficiency and effectiveness. The method provides a powerful alternative to traditional sampling techniques and is applicable to complex, multi-modal distributions.

**Weaknesses:**

1. Equation 46, frequently referenced in Section 2.5, would be more accessible if directly included in that section or if Equation 25 were referenced instead.
1. The computational cost for evaluating the objective from $t=0$ to $T$ seems a lot. An analysis of wall-clock time or time complexity compared to baseline methods would be beneficial.
1. The paper's comparisons with diffusion-based baseline samplers focus on low-dimensional examples (e.g., 2D Gaussian Mixture Model and 10D funnel distribution). However, high-dimensional comparisons are only performed as part of an ablation study. To show a practical scalability, comparisons in a high-dimensional setting would strengthen the argument for the method’s applicability beyond small-scale examples. I think the section 4.3 is showing a possibility to scale, not a powerful performance even in high dimensions.

**Questions:**

1. In Line 342, could you clarify how $(X^{\hat{b}}_t, A^{\hat{b}}_t)$ can be considered independent of $\hat{b}_t(x)$? A detailed explanation would be helpful.
1. Could you provide insight into how the method achieves multi-modal sampling without simulation? Without simulation, it seems challenging to capture information on unexplored modes.

---

> ### Author Response · Authors · 2024-11-25
>
> We thank the reviewer for their positive assessment of our results and for their questions that have helped us improve our presentation.
>
> **Weakness 1: Eq. (25) vs Eq. (46):** Thank you for pointing this out. It is an unfortunate misprint: Proposition 4 with Eq. (25) is *the same* as Proposition 7 with Eq. (46). To help the reader, we re-state the propositions in the appendix, but somehow we made a mistake in restating Proposition 4, and this error percolated back through the text where references to Eq. (46) should have been references to Eq. (15). We have corrected this.
>
> **Weakness 2: computational cost for evaluating the objective:** The PINN objective must eventually be evaluated all the way through $T=1$ (recall that we use $T$ for annealing only). There is indeed a cost in generating the data and computing the time integral, but this is similar to what one also needs to do in every competing methods based on estimating an additional drift, since this drift must be learned for all $t\in[0,1]$. In the revised vesion we will include the wall-clock times of all the approaches we used.
>
> **Weakness 2: Scalability with dimension:** We have obtained new numerical results and more thorough benchmarks of the method. In particular we have added a harder test with benchmarks to compare to using the 50-dimensional Mixture of Student-T distribution studied in the Beyond Elbos paper. We have also verified the performance of the PINN on the $\phi^4$ model (in addition to the existing action matching demonstration), that demonstrate the scalability of our approach as the dimensionality increases (this is 400 dimensions).
>
>
> We hope you find these clarifications and revisions useful. Thanks again for the valuable feedback.

---

> > ### Comment · Reviewer_QP9k · 2024-11-27
> >
> > Thank you for the further clarification. Unfortunately, my curiosity regarding my questions is not resolved, but the other concerns are sufficiently answered. I will conserve my score.

---

> > > ### Author Response · Authors · 2024-11-29
> > >
> > > We apologize for not answering your questions. Let us remedy this:
> > >
> > > **Question 1: independence.** What we mean is that $(X^{\hat b}_t, A^{\hat b}_t)$ can be considered independent of $\hat b_t(x)$ *as far as the differentiation of the loss over the parameters in this velocity is concerned*. This is because the PINN loss is an off-policy objective that can be evaluated using *any* PDF $\hat \rho_t(x)$. As a result taking the expectation and taking the gradient of this objective commute: that is, we can compute its gradient first, then evaluate the expectation. In this second step, we can set $\hat \rho_t(x) = \rho_t(x)$, which amounts to using $(X^{\hat b}_t, A^{\hat b}_t)$ without having to include them in the differentiation step.
> > >
> > > **Question 2: multi-mode sampling.** Because the walkers $X^{\hat b}_t$ move in a time-dependent potential (with transport added), they can find the modes as they appear in $U_t(x)$ (at least if this potential is picked appropriately). Note that the confusion here may be our use of the terminology *simulation-free*: which we say to mean "not having to backpropagate through the solution of the SDE" as would be necessary, e.g. with a KL-type loss. We have changed this accordingly in the text to avoid confusions.

---

### Official Review · Reviewer_XBnC · 2024-10-28

**Soundness:** 3
**Presentation:** 2
**Contribution:** 3
**Rating:** 6
**Confidence:** 4

**Summary:**

This paper proposed NETS which targets sampling from unnormalized probability distributions.
NETS originates from annealed importance sampling, which is widely known to result in biased samples and relies on importance sampling for correction.
NETS views AIS as an SDE augmented with important weights and incorporates a learnable transport drift function $b(x)$ to correct this biasedness.
This idea is similar to the stochastic control.
The core ingredient of NETS, $b(x)$, can be learned by PINN or action matching, and in the first case, the learning is off-policy and does not require backpropagating the SDE.
The alternative perspective on AIS is very interesting and insightful, and improving it with an additional transport map seems novel and effective in practical applications.

**Strengths:**

- To the best of my knowledge, the proposed method is novel and firstly tackles the biasedness problem of AIS with an additional transport function. The weight-augmented SDE formulation is also inspiring for follow-up works.
- The authors derive two practical training methods for the transport function $b$. Especially, for the action matching objective, computing the Hessian matrix of $\phi_t(x)$ can be avoided by a smart trick on $A_t^b$.
- The experiments are illustrative and well-designed to demonstrate the superiority of NETS. I especially favor the illustration in Figure 1 that shows the effectiveness of the transport.

**Weaknesses:**

- NETS is somewhat similar to stochastic control for me. I suggest the authors clarify the similarities and differences between them.
- There are some critical hyperparameters that can affect the performance of NETS, e.g., the $\varepsilon_t$ parameter and the stepsize of discretization. I would like to see how these parameters affect the convergence of the proposed method. Besides, it would be better for the authors to give some instructions on how to select these hyperparameters in practice.
- Although the authors target sampling tasks, an important application of NETS is to calculate expectations based on equation (21). Exploring or discussing this application would be interesting.
- For methods that require simulating the trajectories or computing divergence/hessian, computational cost is always a major concern. The authors should report the training time and memory consumption of NETS and other baselines, to fairly compare different methods.
- There are many typos in mathematical equations that can cause misunderstanding. I strongly suggest the authors carefully check their manuscript.
    - The expression of $\rho_t(x)$ in equation (9).
    - The reference to equation (9) in line 193 seems incorrect.
    - The expression of the RHS in equation (18).
    - The expression of $g_t$ in equation (40).
    - 'Them' in line 38; 'resut' in line 312; and so on.
- The authors give a guarantee of convergence in KL divergence. However, no KL divergence results have been compared or reported in the experiments.

**Questions:**

- For the PINN loss in equation (25), we need to compute the divergence of $\hat{b}_t(x)$ to obtain the gradient. Then how do you efficiently compute it, especially in the high-dimensional case?

---

> ### Author Response · Authors · 2024-11-25
>
> We thank the reviewer for their questions and suggestions, which we have tried to address next. If you find these clarifications and revisions useful, we would appreciate it if you improved your score.
>
> **Weakness 1: comparison with methods based on SOC:** The problem of sampling a target distribution can indeed be tackled via the solution of a stochastic optimal control (SOC) problem  -- this is what is proposed e.g. in Ref. [1] (which we cite) where the authors introduce a method to estimate a Föllmer process, i.e. a Schrödinger bridge between a point mass distribution and the target. The main difference is that, in the SOC formulation, the control = drift in the SDE must be learned from scratch, which is typically challenging since the reference process (i.e. the SDE without the control) has solutions whose law at final time is far from the target distribution in general. In contrast, our approach can be viewed as a **guided search** in which we predefine the path in distribution space between the base and the target. The resulting process may not be optimal in the sense of SOC, but the learning of the drift is facilitated in our formulation. In addition it allows us to choose any suitable base suitable distribution, whereas the SOC formulation used in Ref. [1] leads to a Föllmer process whose base distribution must be a point mass. This too limits the flexibility/scalability of this approach compared to ours.
>
> [1] Zhang, Q. & Chen, Y. (2022). Path Integral Sampler: a stochastic control approach for sampling. arxiv preprint arXiv:2111.15141.
>
> **Weakness 2: choice of hyperparameters:** the main metaparameters to choose in the training procedure are: (i) the annealing schedule controlled by $T$ in the loss, and the choice of number of discretization points. Both can be adjusted on-the-fly by monitoring the ESS, and ensuring that it does not deteriorates as the annealing proceeds. This is what we did in all of our experiments: if the annealing is too agressive, the training has a hard time to converge. We explain this point better in the revised version.
>
> Regarding the diffusion coefficient $\epsilon_t$, it is important to stress again that it can be adjusted post-training. Theoretically, the $\epsilon_t \rightarrow \infty$ limit is perfect sampling. We demonstrate in Figure 2 that this limit can be more practically realized *when you include transport* than without.  We  also discuss this point in more details in the revised version.
>
> **Weakness 3: calculation of expectations:** The main purpose of NETS is to calculate expectations as well as estimate the partition function/Bayes factor $Z_1 = \int e^{-U_1(x)} dx$ (which is an important quantity in practice). In the paper we focus on $Z_1$ (as is also done in many of the sampling papers cited), mostly because this factor allows us to benchmark the accuracy/efficiency of our method. For the $\phi^4$ example we compare expectations via analysis of the magnetization given in the figure.
>
> **Weakness 4: computational cost:**
> Training times are drastically influenced by how good a coder you are :), so we do not think they are a great scientific metric. But for comparison it took about 23 minutes for the 40-mode GMM to train. Compared to the ESS and $W_2$ metrics quoted from iDEM, this is fast. But again, they may have just coded things differently, and we do not feel that it is fair for us to claim something here.
> Please see the discussion in the appendix on the Hutchinson trick, e.g. for dealing with the cost of the divergence.
>
> **Weakness 5: typos:** Thank you for pointing these out. We have corrected them and have gone through a careful read of our paper to correct a few more.
>
>
> **Question: divergence calculation:** In practice this divergence can be efficiently calculated using Hutchinson's trace estimator with antithetic variates (similar to what is used to estimate the divergence of the score in score based diffusion models). More specifically, we can use
> $$
> \nabla \cdot b_t(x) = \frac1{2\delta} \mathbb{E}  \big[ \eta \cdot\big(b_t(x+\delta \eta )- b_t(x-\delta \eta ) \big)\big] + O(\delta^2)
> $$
> where $0<\delta \ll1$ is an adjustable  parameter and $\eta \sim N(0,\text{Id})$. This estimator can be made unbiased by using two independent copies of $\eta$ for the terms involving the square of $\nabla \cdot b_t(x)$ in the loss. We tested it on some of the examples and didn’t seem to make much of a difference in final performance as compared direct calculation. If the reviewer would like we could include some of these numbers in a camera-ready version.

---

> > ### Comment · Reviewer_XBnC · 2024-11-28
> > **Feedback to the authors**
> >
> > Thank you for your response to my concerns. However, my weakness 6 regarding the KL divergence is not addressed by the authors. It would be good if this result aligned with the theoretical analysis. As such, I will maintain my current score.

---

> > > ### Author Response · Authors · 2024-12-01
> > >
> > > Thank you for this comment.
> > >
> > > Unlike estimating the PINN loss, computing the KL is challenging in general, which is precisely why the bound we derive is useful. Nevertheless we can verify it numerically for GMM models: the result of this computation aligns with the theoretical analysis. It can be found at:
> > >
> > > https://drive.google.com/file/d/1O7ATPsloqZObjaMET0qkGhRPx8GbsIB0/view?usp=sharing
> > > https://drive.google.com/file/d/1m51Xj57IxexEb-warb6O1uiWb6JLp0bQ/view?usp=sharing
> > >
> > > In addition, the reviewer may find the following calculation useful. Assume that the base distribution is a $N(0,Id)$ and the target is a $N(b,Id)$ with some $b\in \mathbb{R}^d$. Assume also that we take
> > > $$
> > > U_t(x) = \frac12 |x-bt|^2
> > > $$
> > > If the learned drift is some constant $\hat b\in \mathbb{R}^d$ not necessarily equal to $b$, an explicit calculation shows that the solution to $\partial_t \hat \rho_t + \nabla \cdot(\hat b \hat \rho_t)=0$, $\hat \rho_{t=0}=\rho_0$ satisfies
> > > $$
> > > D_{KL}(\hat \rho_{t=1}\|\rho_1) = \frac12 |\hat b-b|^2
> > > $$
> > > whereas the PINN loss is in this case given by
> > > $$
> > > L^{T=1}_{PINN}(\hat b,F) = |\hat b-b|^2 + \frac13 |\hat b- b|^4
> > > $$
> > > confirming that $D_{KL}(\hat \rho_{t=1}\|\rho_1) \le \sqrt{L^{T=1}_{PINN}(\hat b,F)}$ with equality *iff* $\hat b = b$. (Note that for small $|\hat b - b|$, $\sqrt{L^{T=1}_{PINN}(\hat b,F)} \sim |\hat b- b|$.)
> > >
> > > We hope that this now fully addresses your concerns, and if so, would appreciate any increase in score.

---

> > > > ### Comment · Reviewer_XBnC · 2024-12-02
> > > >
> > > > My concerns are addressed, and thus I have increased my score by one point.

---

### Official Review · Reviewer_z7G3 · 2024-11-02

**Soundness:** 3
**Presentation:** 4
**Contribution:** 3
**Rating:** 6
**Confidence:** 4

**Summary:**

This paper proposes a sampling algorithm using annealed Langevin dynamics and additional learnable transport, which can be viewed as a variant of annealed importance sampling. The additional learnable transport can be used to reduce the high variance appeared in original annealed importance sampling. The paper proposes to learn the additional transport by minimizing two objective functions. Theoretically, they show the PINN objective controls the KL divergence between the model and its target. Last, they provide numerical examples outperforming several baseline state-of-the-art techniques and testify the scalability of the method.

**Strengths:**

- The idea of combining annealed importance sampling and additional transport is interesting and novel.
- The theoretical result of the KL control for the PINN objective is interesting.
- Considering both $\hat{b}_t$ and $\hat{F}_t$ simultaneously in the PINN objective is interesting and novel.
- This method contains correction for sampling errors if the additional transport is imperfectly learned.

**Weaknesses:**

-  In practice, using PINN objective need compute the divergence of a velocity field in $\mathbb{R}^d$, which can be extremely inefficient in high-dimensional case.
-  Several studies have investigated sampling algorithms by solving the velocity field through partial differential equations ([1], [2]). Therefore, a more detailed discussion of these related works would enhance the author’s contribution by clarifying connections and advancements in this area.
-  Although the PINN objective is off-policy, making it more computationally efficient, the performance of the algoirthm can be sensitive to the choice of $\hat{\rho}_t$. For example, it may lead to poor exploration when $\hat{\rho}_t$ is significantly far away from $\rho_t$.  A more detailed discussion on this trade-off would strengthen and complete the author’s analysis in this context.

References

1. Tian Y, Panda N, Lin Y T. Liouville Flow Importance Sampler[J]. arXiv preprint arXiv:2405.06672, 2024.
2. Fan M, Zhou R, Tian C, et al. Path-Guided Particle-based Sampling[C]//Forty-first International Conference on Machine Learning.

**Questions:**

see weaknesses.

Minor comments: typo in Line 140-142: $\partial_t \log F_t$ should be $\partial_t F_t$.

---

> ### Author Response · Authors · 2024-11-25
>
> We thank the reviewer for their questions and  suggestions. Our answers follow - please let us know if there is any other question we could address in order for you to raise your score.
>
> **Weakness 1: need to compute the divergence in the PINN loss:** In practice this divergence can be efficiently calculated using Hutchinson's trace estimator with antithetic variates (similar to what is used to estimate the divergence of the score in score based diffusion models). More specifically, we can use
> $$
> \nabla \cdot b_t(x) = \frac1{2\delta} \mathbb{E}  \big[ \eta \cdot\big(b_t(x+\delta \eta )- b_t(x-\delta \eta ) \big)\big] + O(\delta^2)
> $$
> where $0<\delta \ll1$ is an adjustable  parameter and $\eta \sim N(0,\text{Id})$. This estimator can be made unbiased by using two independent copies of $\eta$ for the terms involving the square of $\nabla \cdot b_t(x)$ in the loss. We have added a discussion of this in the appendix in the implementation section. We have tested that this approach works in practice and it seems to work fine. If the reviewer likes, we can add experimental infor for this to a potential camera-ready version.
>
>  **Weakness 2: more detailed discussion of related works:** The results in the paper by Tian et al. are indeed very relevant to ours, as already mentioned in our original submission. With respect to this work, our results bring several important novelties:
>
> On the theoretical side, we introduce several losses to estimate the drift, and show that the PINN loss controls  the variance of the variance of the weights as well as the KL divergence. We also show that the method can be applied to Langevin equation with time-dependent diffusion coefficient $\epsilon_t$, even though the additional drift is the same for all $\epsilon_t \ge 0$ - in contrast the paper by Tian et al. focus on the probability flow ODE obtained when $\epsilon_t=0$). Finally, in the revised version, we show that the method can be generalized to the overdamped non-equilibrium dynamics (i.e, adding inertia) and can also be used to sample distributions that vary across more parameters than the single scalar $t$.
>
> On the numerical side, we show that adjusting the diffusion coefficient (and making it non-zero in general) is key to improve performance, both at learning and sampling stages. For the latter, this adjustment can be done post-training. We also demonstrate the performance of our method in a larger class of examples.
>
> The method in the paper by Fan et al. also aims at constructing a probability flow ODE ($\epsilon_t=0$). As far as we could tell, the results in this paper are not fundamentally different from those in the paper by Tian et al.
>
> We will add a more detailed discussion of these works in the revised version, to highlight the difference with our proposed approach.
>
> **Weakness 3: sensitivity of the PINN loss to the choice of $\hat{\rho}_t$.** Indeed, in practice the choice of this reference density matters, and that is why we always aim to use $\hat{\rho}_t = \rho_t$. The key point, however, is that, *since the PINN objective can be used off-policy, its performance are less affected by the errors we make in the sampling (i.e. by the variance of the weights used to sample wrt $\rho_t$) than, say, those of the AM loss (which must be used on-policy).* We will add a more careful discussion in the revised version to stress this point.

---

> > ### Comment · Reviewer_z7G3 · 2024-11-26
> >
> > I would like to thank the authors for their detailed response. As I already voted for acceptance, I will keep my score.

---

### Official Review · Reviewer_dGdu · 2024-11-03

**Soundness:** 2
**Presentation:** 2
**Contribution:** 1
**Rating:** 5
**Confidence:** 4

**Summary:**

This work presents an approach to sample from densities specified up to a normalizing constant using controlled annealed Langevin dynamics. The approach is first motivated by Jarzynski's equality, which can be interpreted as a time-continuous variant of annealed importance sampling (AIS). Then, the paper extends this framework to *controlled* Langevin dynamics and proposes to learn the drift either by physics-informed neural networks (PINNs) or a version of action matching. The resulting method, called Non-Equilbrium Transport Sampler (NETS), is numerically evaluated on Gaussian mixtures, the Funnel distribution, and the simulation of a lattice field theory.

**Strengths:**

The paper is generally well-structured and theoretically sound. In view of existing work (see weaknesses below), the paper provides the following contributions:
1. New (and arguably simpler) derivation of existing results on (controlled) annealed Langevin dynamics,
2. Novel objective based on action matching (AM),
3. Evaluation of additional tricks, such as usage of importance weights for the PINN objective, curriculum learning for the time interval, and combination with resampling.

**Weaknesses:**

**Related works:** Large parts of the paper are already included in existing work:

- *Controlled Langevin dynamics:* While Vargas et al. (2024) are mentioned in the related works, it is important to note that the *main* proposition of the present paper, Proposition 3 (Nonequilibrium Transport Sampler (NETS)), is already presented in Vargas et al. (2024, Proposition 3.3). Using the data processing inequality, the latter result also seems to yield the bound on the KL divergence in Proposition 5.

- *PINN objective:* The PINN objective in Section 2.5 is already stated in https://arxiv.org/abs/2301.07388 (Section 3, "The continuity loss"), https://arxiv.org/pdf/2405.06672 (Section 2.2), and https://arxiv.org/pdf/2407.07873 (Section 3.2, "Annealing"), where also on- ("uniform") and off-policy ("along trajectories") methods for $\hat{p}_t$ are explored. While both sampling variants do not require backpropagating through the SDE integrator, the latter *does* need to simulate the SDE (at least periodically if using a buffer). Thus, I would not say that NETS-PINN is "simulation-free even if used on-policy," as stated in line 344.

Because of these observations, the present work only provides little *novel* contributions (see strengths above). In particular, the PINN objective seems to outperform the AM objective in the first two experiments, and curriculum learning is a relatively standard trick for time-dependent PINNs.

Moreover, many of the shortcomings mentioned in related works are either not completely accurate or have already been tackled. In particular, there seem to be several methods having checkmarks for all columns of the table on page 2. Some details (see also questions below):

1. There exist several off-policy losses for samplers based on dynamical transport, which, in particular, do not require backpropagating through the SDE integrator and lead to unbiased sampling methods; see (sub-)trajectory/detailed-balance losses for GFlowNets (e.g., https://arxiv.org/abs/2210.00580, https://arxiv.org/abs/2310.02679, https://arxiv.org/pdf/2402.05098) and log-variance (LV) losses (https://arxiv.org/abs/2307.01198; also used in Vargas et al. (2024)). In particular, these losses also fall into the category of "optimize-then-discretize".

2. I assume the LV loss is meant in the second part of the sentence "this objective either needs backpropagating through the SDE or has to be computed with a reference measure which may make it high variance". However, for the typical choice of using the on-policy drift, the reference measure is actually *reducing* variance (as compared to the KL divergence) (https://arxiv.org/abs/2307.01198, https://arxiv.org/abs/2005.05409).

3. The fact that the objectives of DDS (and DIS, see Berner et al., 2024) can also be viewed as SOC problems shows that methods based on SOC do *not* need to start with samples from a point mass.

---

**Experiments:** Important baselines and ablations seem to be missing:

1. CMCD: As argued above, CMCD has the same theoretical framework, and it can also be viewed as a BSDE (instead of PINN) loss, as shown in https://arxiv.org/pdf/2407.07873 (Prop. 3.1). However, NETS is currently not compared against CMCD.

2. ODE importance weights: The advantage of using the SDE instead of the ODE during generation should be validated. Specifically, one could also use the learned vector field $\hat{b}$ to simulate the ODE (aka continuous-time normalizing flow) and compute the importance weights using the change-of-variables formula (as is done in some of the works mentioned above that also use the PINN objective).

3. Sections 4.3 and 4.4 do not provide any baselines except for AIS and only evaluate one of the proposed NETS versions. Moreover, no details on the AIS settings seem to be provided.

4. Since NETS seems to require a very high number of discretization steps (all experiments use >= 200 and some up to 1500-2000 steps), it would be interesting to see how well NETS is working for fewer steps (which is relevant in settings where target evaluations are expensive). Is the same number of steps also used during training?

5. More benchmarks (e.g., from Blessing et al. (2024)) are required to judge the performance of NETS against other state-of-the-art samplers, and it would be good to see an ablation of some components of NETS (e.g., the curriculum learning or the importance weights in the PINN objective).

When quoting baseline results from other works, it is crucial to follow the same evaluation protocol, in particular, the number of integrator steps and evaluation samples:

1. *Funnel:* NETS: 200 steps and 10000 samples. Baseline: 128 steps and 2000 samples.
2. *Gaussian Mixture:* NETS: 250 steps and 2000 samples. Baseline: 100 steps (at least for PIS; it might also be that ESS is computed using an "exact likelihood from a continuous normalizing flow (CNF) model [trained] using optimal transport flow matching on samples from each model," which "keeps the number of function evaluations per integration around 100 in practice") and 1000 samples.

Please reevaluate your results using the same protocol. Finally, it would be interesting to also compare training costs, e.g., time and number of target (& gradient) evals, etc.

**Questions:**

1. It is unclear why the paper writes that off-policy objectives do not need samples from the "target density" (line 63) since we do not seem to have samples from the target density for both on- and off-policy losses. In this context, "off-policy" typically means that the trajectories do not need to follow the ones from the *learned* model.

2. What exactly is meant by "grid-free learning," i.e., why are PIS and NETS grid-free but DDS and CMCD are not? All these methods are based on simulating SDEs on some time grid. In particular, for off-policy methods, the time grid can basically be chosen arbitrarily.

3. It is unclear why DDS is only "partially" unbiased. For diffusion-based samplers, one can perform importance weighting in path-space, which is unbiased (up to bias introduced from self-normalization as also used in NETS).

4. What prevents other samplers based on diffusion models (such as DDS and DIS) from using "arbitrary step size and time-dependent diffusion after learning the model"? As already shown in https://arxiv.org/abs/2106.02808, there is a family of equivalent SDE that can be used for inference and which only requires knowledge of the score. Naturally, the approaches might overfit to a fixed time discretization. However, the time steps can be randomized during training, as is done in NETS.

5. How does the last step in (52) follow (since $\hat{b}$ is not at the optimum)?

---

**Suggestions:** While the largely algebraic calculations might yield easier proofs of existing results, the paper would benefit from providing further explanations and context. Two examples:

- The sentence "The effect of the last term at the right hand-side of this equation can be accounted for by using weights." only becomes clear after reading Remark 1. It might generally be good to add some more intuition to these algebraic calculations, e.g., (5) is just showing that $\rho_t$ is the invariant measure of the Langevin dynamics (with noise $\sqrt{2\varepsilon_t}$).

- One could also mention that Eq. (13) is just the continuity equation with predefined density evolution, and Eq. (14) are the FPEs of the corresponding family of SDEs with the same marginals (by standard manipulations of FPEs, e.g., Appendix G in https://arxiv.org/pdf/2106.02808). This also directly implies the statement of Prop. 2.

See weaknesses above for additional suggestions and questions.

---

**Typos:**

- $R^d$ on line 122

- $\partial_t \log F_t$ on line 140

- $R^{d+1}$ in line 160

- $U$ (without time index) in (10) and line 123

- 9 in line 183 should probably be (8)

- 46 in lines 272 and 275 should probably be (25)

- THe in line 852

- These in line 971

- There seem to be $\hat{cdot}$ missing over the $\phi$ in Section 2.6 and the corresponding proofs.

- closing $)$ missing in (18).

- it should be integration against $\hat{\rho}_t$ at several places in the proof of Prop. 5.

---

> ### Author Response · Authors · 2024-11-25
>
> We thank the reviewer for the detailed and itemized review that has helped us greatly to contextualize our contributions and improve our presentation. Below we give an itemized response to each of your theoretical and experimental remarks. We hope that this answer will allow us to make our contributions clear to you and that you will consider raising your score as a result.
>
> *__Related works__*
>
> > Controlled Langevin dynamics (Vargas et al. 2024):
>
> We have corresponded extensively with the authors of this work, both to properly benchmark their approach as well as highlight its differences with our method. Please note that the essential machinery of Proposition 3.3 in Vargas et al. has been known for 20 years in the statistical physics community (Jarzynski 1999, Vaikuntanathan & Jarzynski 2008), and neither us nor Vargas et al. are claiming to have discovered this equality. However, we both provide different derivations of it that make it more interpretable for sampling. Vargas et al. prove this result through the use of Girsanov, and here we provide a proof of it through simple manipulations of the the Fokker-Planck equation and other PDEs. This uniquely allows us to:
> - Directly relate a PINN loss to the Jarzynski weighting factors.
> - Show that we can better approach $\epsilon_t \to \infty$ limit in practice. Note that for our setup, perfect sampling is achieved in this limit, whether the learned transport is perfect or not. While this limit cannot be reached in practice without transport (as it would require taking astronomically large value of $\epsilon_t$ in general), we show that, with some learned transport added, even moderate values of $\epsilon_t$ can improve the sampling dramatically. This feature can be exploited after training as an explicit knob for tuning performance vs cost. This has not been recognized in other ML sampling literature nor in Vargas et al.
> - Incorporate SMC resampling into the generation process which allows for increased performance (see both tables), which is also not realized in Vargas et al.
> - Directly control the KL-divergence with the PINN loss.
> - Directly characterize the minimizer of the action matching loss, which is also not known from previous work on it.
>
>
> We would like to stress that in  our exchanges with authors of Vargas et al. they have explicitly differentiated our work from theirs with these points and suggested we make these delineations clear in this rebuttal. We are glad that you can discern a connection to our KL-bound through more analysis to a proposition in their work, but would like to stress that it is a non-trivial result in general, and a wider audience who is less familiar with this topic would not intuit this result as it is not proven anywhere except in this work.
>
>
> > PINN objective: The PINN objective in Section 2.5 is already stated in...
>
> We appreciate your help in finding these related works and citations, and we have now incorporated them directly into the related works of the text. Please let us know if you think we can improve these citations further. One of these works seems to be essentially co-incident as it only appeared on the arxiv 2 months before this submission.
>
> We would like to stress that this PINN loss was not known to be intrinsically connected to the Jaryznski equation and annealed langevin dynamics, which we elucidate here. This fits in naturally with physical interpretation of this equality -- it is a direct control on the variance of the Jarzynski weights (the dissipation) in the process connecting $\rho_0$ to $\rho_1$. We have added an additional statement to this in the text.
>
> > Thus, I would not say that NETS-PINN is "simulation-free even if used on-policy," as stated in line 344
>
> This is a misalignment of the meaning of simulation-free on our part, which we say to mean "not having to backpropagate through the solution of the SDE" as would be necessary, e.g. with a KL-type loss. We have changed this accordingly in the text.
>
> > Moreover, many of the shortcomings mentioned in related works ...
>
> We have entirely removed the table because we realize some of the column headings drifted from our original meaning. For example, when we said unbiased, we meant very specifically that the importance weights along the dynamical transport *even* correct discretization errors in the SDE. We have included an appendix deriving this result (Sec 5.2). Thanks for pointing us to a few other citations, which we have now included in the related works.

---

> > ### Author Response · Authors · 2024-11-25
> >
> > > I assume the LV loss is meant in the second part of the sentence...
> >
> > By variance we did not mean the variance of gradients of the objective, we meant it more colloquially. After conferring with a CMCD author, we both agreed on the phrase "numerically unstable". Indeed, in our new benchmarks against CMCD that uses the LV loss (detailed below), we observe that CMCD would NaN without a critical number of bridge steps, strict clipping of the model gradients, strict clipping of the target log densities gradients, early stopping of the training, and a careful implementation of the interpolation schedule. We have adjusted to the text to say this, and make no claims about the theoretical control of the variance on the objective, which we think is elegant.
> >
> >
> > *__Experiments__*
> >
> > > Comparison to CMCD:
> >
> > In careful correspondence with  authors of CMCD, we have now benchmarked their method on the targets already in this paper as well as a new 50-d Mixture of Student-T distribution. We have used the exact same benchmarking code as them with the same hyperparemers values that you had asked about (number of MMD samples, etc). The LV loss only worked with 256 time steps, so we used that and 100 for ours throughout as you had asked. We note that *both the PINN and the action matching loss outperform CMCD in all experiments*.
> >
> >
> > > ODE importance weights: The advantage of using the SDE instead of the ODE during generation should be validated.
> >
> > We also found this to be an interesting question, but we hope to think we have already validated it in Figure 2 (if we understand your question correctly). This is where we theoretically verify the following fact about annealed langevin dynamics (*with or without learned transport included*): As $\epsilon_t \to \infty$, ESS $\rightarrow 1$ (perfect sampling). This is a fact arising from how the ESS is defined in terms of the Jarzysnki weights; and interestingly, the PINN loss directly controls these weights (the ones coming from annealed langevin dynamics with transport). In general, the limit $\epsilon_t\to\infty$ cannot be reached in practice without transport (as it would require taking astronomically large value of $\epsilon_t$ in general), but we show that, with some learned transport added, even moderate values of $\epsilon_t$ can improve the sampling dramatically. The left-most datpoints in Figure 2 are the ESS without diffusion (i.e. ODE sampling as you had asked).
> >
> > We further demonstate this phenomena under the $W_2$ metric for the action matching model trained on the MoS target. That's given in the appendix. This is an essential feature of our presentation.
> >
> > > $\phi^4$ evaluation
> >
> > We have now benchmarked the PINN loss on the target distributions from $\phi^4$ models and shown that it performs equivalently to the AM loss. We will include this in the figure by Tuesday night once a work maintenance ends on the cluster used to produce it Monday morning (this is where the data for the plot sits). The AIS setup is NETS without transport, and we have now tried to clarify that in the text.
> >
> > > Since NETS seems to require a very high number of discretization...
> >
> > NETS does not necessarily need a high number of discretization steps. This number was not chosen against any metric in the first draft. All experiments in the code now use 100 steps, except the $\phi^4$ example, which is intrinsically stiff due to the phase transition (though this could be potentially alleviated by sampling in Fourier space). We use 100 steps during training.
> >
> > > More benchmarks
> >
> > As mentioned, we have now included an additional benchmark on the 50-d Mixture of Student distribution from the paper you suggested, and shown that NETS strongly performs with both objective functions on this example. In addition, we have included a measure of $W_2$ distance in accordance with Blessing et al. for all experiments, and made sure the implementation of evaluation metrics were equivalent for each table.
> >
> >
> > > Training times
> >
> > Training times are drastically influenced by how good a coder you are :), so we do not think they are a great scientific metric. But for comparison it took about 23 minutes for the 40-mode GMM to train. Compared to the ESS and $W_2$ metrics quoted from iDEM, this is fast. But again, they may have just coded things differently, and we do not feel that it is fair to claim something here.

---

> > > ### Author Response · Authors · 2024-11-25
> > >
> > > *__Question__*
> > >
> > > > It is unclear why the paper writes that off-policy objectives do not need samples from the "target density"
> > >
> > > By off-policy, we mean that the expectation in the objective need *not* be taken with respect the true $\rho_t$. By on-policy, we mean that this expectation *does* need to be taken with respect to the true $\rho_t$ (which can be done with importance weights). Please let us know if you use this terminology differently.
> > >
> > > > What exactly is meant by "grid-free learning,"
> > >
> > > We mean that we learn $b_t(x)$ or $\phi_t(x)$ globally for all $t \in [0,1]$, and not on a fixed grid. While one could argue that other methods that we compare to could do this in retrospect, they do not in their presentation nor their experiments. Here, we stress this as a feature because it allows us to vary the time-discretization after training (which is useful if you want to change your diffusion coefficient for increased performance!)
> > >
> > > > It is unclear why DDS is only "partially" unbiased.
> > >
> > > To quote above, we have removed this table entirely because of ambiguity by what we meant by bias. Thanks for catching this.
> > >
> > > > What prevents other samplers based on diffusion models from using arbitrary diffusion coeff?
> > >
> > > We agree this is possible, but want to again stress that the way we learn here is to exploit a fact that we have about annealed langevin dynamics with or without transport: $\epsilon \rightarrow \infty$ is perfect sampling.
> > >
> > > > How does the last step of (52) work out?
> > >
> > > The last step in Eq. (52) was incorrect, and so was the result in Proposition 5, thank you for pointing this out. We have modified the statement of the proposition as well as its proof.
> > >
> > >
> > >
> > >
> > >
> > > **Thank you for your thorough feedback.** We really appreciate it and think it has strongly improved our paper.  If there is anything else we can address, please let us know.
> > >
> > > 1. Jarzynski et al (1996) *A nonequilibrium equality for free energy differences*
> > > 2. Vaikuntanathan et al (2007) *Escorted Free Energy Simulations: Improving Convergence by Reducing Dissipation*
> > > 3. Blessing et al (2024) *Beyond ELBOs: A Large-Scale Evaluation of Variational Methods for Sampling*

---

> > > > ### Author Response · Authors · 2024-12-01
> > > >
> > > > Thank you again for your detailed comments. We took great effort in addressing them and revising our paper accordingly. To do so we also had extensive discussions with the authors of the CMCD paper who helped us delineate the difference between our works (and pointed out themselves several novel aspects of our method). Since the discussion period ends soon, we are kindly asking you for feedback on our rebuttal.

---

> > > > > ### Comment · Reviewer_dGdu · 2024-12-02
> > > > >
> > > > > Thank you for your extensive revision and response. Generally speaking, I think that both the presentation and empirical evidence have improved. However, the amount of experiments (only 3 targets with a sufficient number of baselines) is still less than typical papers in the field, in particular, given that large parts of the theory have already been known (see also my responses below).
> > > > >
> > > > > Apart from several comments (see below), I have two theoretical questions:
> > > > > 1. $\epsilon_t$-limit: Could you please elaborate on the $\epsilon_t \to \infty$ limit? It seems not obvious from Prop. 3 in your work, how this limit would be well-defined. Moreover, "perfect sampling" with "non-perfect learned transport" is also possible outside of this limit by using the appropriate weights as shown in Prop. 3. In particular, how does this differ from picking an SDE with the same marginals (one of which is called probability flow ODE) during sampling, which can be done for any diffusion-based sampler where the score is known.
> > > > > 2. ODE importance weights: It seems that the weights for $\epsilon=0$ in Proposition 3 do not correspond to how the weights are computed for normalizing flow (i.e., via the change-of-variables formula, i.e., integrating the divergence of the drift along the trajectories; see, e.g., https://www.jmlr.org/papers/volume22/19-1028/19-1028.pdf, Eq. (81)). However, this would also lead to "perfect sampling" for any (non-perfect) vector field $\hat{b}$ learned via the PINN or AM objective.
> > > > >
> > > > > **Related works**
> > > > > > CMCD
> > > > >
> > > > > I agree that the essential machinery of Proposition 3. has been known, and as written in my review, I acknowledge the new and arguable simpler derivation. However, I do not yet see why your work would bring "unique" benefits or make it more "interpretable" for sampling:
> > > > >
> > > > > 1. While I agree that Vargas et al. (2024) do not explicitly write down the PINN loss, the relation between the PINN loss and the weighting factors is already given in Vargas et al. (2024, Proposition 3.3).
> > > > > 2. While SMC has not been used for CMCD yet, SMC steps have been leveraged for sampling with diffusion models in PDDS (https://arxiv.org/pdf/2402.06320), which is neither mentioned nor compared against.
> > > > > 3. As written in my response, the data processing inequality shows that also the loss in Vargas et al. (2024) allows to control the KL-divergence.
> > > > > 4. As written in my review, I agree that the AM objective is novel in the context of sampling. However, the fact that the minimizer of the action matching loss corresponds to the unique gradient field satisfying the corresponding continuity equation seems to be derived in the "action matching" paper (https://arxiv.org/abs/2210.06662).
> > > > > 5. It seems that CMCD, in combination with the log-variance loss, also gives control over the Jarzynski/importance weights.
> > > > >
> > > > >
> > > > > > PINN objective
> > > > >
> > > > > Note that the work appearing 2 months before the submission seems to have already presented the objectives in March in https://openreview.net/forum?id=KwHPBIGkET.
> > > > >
> > > > >
> > > > > > Simulation-free
> > > > >
> > > > > Thank you for adapting the term "simulation-free" and removing the problematic table. Note that backpropagating through the solution of the SDE can also be prevented for KL-based losses using a "optimize-then-discretize" approach via the adjoint SDE.
> > > > >
> > > > > > Discretization errors
> > > > >
> > > > > Thank you for providing an additional analysis in Section 4.2. However, note that also the weights derived in CMCD hold in discrete time (i.e., account for the discretization errors in the SDE).
> > > > >
> > > > > > LV
> > > > >
> > > > > While it might be numerically unstable in certain settings, it seems that the LV and trajectory-balance loss can achieve strong performance (see https://arxiv.org/abs/2402.05098 and https://arxiv.org/abs/2307.01198 as mentioned in my review). Moreover, it should at least be mentioned that *in theory* also competing methods do not rely on a "fixed grid" and could be used with randomized time steps.
> > > > >
> > > > > **Experiments:**
> > > > >
> > > > >
> > > > > > CMCD, $\phi^4$
> > > > >
> > > > > Thank you for adding experiments. However, it seems that CMCD has only been evaluated for GMM and Funnel, but not for MoS or the $\phi^4$ experiments? I also could not find the NETS-PINN results for the latter target in the current version.
> > > > >
> > > > > > Training times
> > > > >
> > > > > I think that good code is also a contribution to the scientific community.
> > > > > In particular, for sampling problems (where one does not rely on data and can potentially train infinitely long), it seems crucial to also consider the training time.
> > > > >
> > > > > **Questions:**
> > > > >
> > > > > > Off-policy
> > > > >
> > > > > Typically, see, e.g., https://arxiv.org/abs/2402.05098, "off-policy" refers to using a policy that is different from the "current" policy (as opposed to the "target" policy), i.e., in your case it would be the density given via the model during optimization instead of the target annealed density.
> > > > >
> > > > > > Prop. 5
> > > > >
> > > > > Thank you for fixing the result.

---

> ### Author Response · Authors · 2024-12-03
>
> We thank the reviewer for their reply. Below we provide answers to your questions and address your follow-ups. Please note that we cannot edit the PDF at this point until the potential camera-ready version, but we reference simple changes we would make to it in such a case:
>
> **Question 1 ($\epsilon_t \to \infty$ limit):** The limit as $\epsilon\to\infty$ is well-understood: in this limit, the walkers evolve infinitely fast compared to the potential, and as a result are always in quasi-equilibrium with it, so that the weights become unnecessary -- this observation can be made precise using standard limit theorems for diffusions evolving on multiple time scales (see e.g. the book by Pavliotis & Stuart referenced as [1] below), and it is at the basis of thermodynamic integration strategies.  Of course, the integration of the SDE must be done on the fast time scale $\tau = \epsilon t$, i.e. the computational cost increases with $\epsilon$. The question therefore becomes *whether we can reach the limit $\epsilon\to\infty$ in practice* (i.e. whether we can work with values of $\epsilon$ large enough that we are effectively in this limit).  Without transport, this is not possible in general -- the limit is only attained for values of $\epsilon$ that are astronomically large (e.g. because of mestastability effects). However *our numerical results show that the situation can change dramatically with some transport added, in which case increasing $\epsilon$ do increase the ESS significantly* (see Fig. 2 for illustration).
>
> **Question 2 (ODE weights):** The weight computation that we use is a generalization of the one obtained with the probability flow ODE (when $\epsilon_t=0$). This can be seen as follows:
>
> Suppose that we evolve the walkers using the probability flow ODE
> $$
> \dot X_t = \hat b_t(X_t), \qquad X_{t=0} = x_0 \sim \rho_0
> $$
> with an *imperfect* drift $\hat b_t(x)$. In this case the importance weights to use are
> $$
> \frac{\rho_1(X_{t=1})}{\hat \rho_{t=1}(X_{t=1})}
> \equiv \frac{e^{-U_1(X_{t=1})+F_1}}{\hat \rho(X_{t=1})}
> $$
> where $\hat \rho_t(x)$ is the solution to the PDE
> $$
> \partial_t \hat \rho_t + \nabla \cdot (\hat b_t \hat \rho_t )=0, \qquad \hat \rho_{t=0} = \rho_0 \equiv e^{-U_0 + F_0}
> $$
> This equation can be solved by the method of characteristics, which shows that
> $$
> \hat \rho_{t=1} (X_{t=1}) = \rho_0(x_0) \exp\left( - \int_0^1 \nabla \cdot \hat b_t (X_t) dt\right)
> $$
> i.e.
> $$
> \hat \rho_{t=1} (X_{t=1}) = \exp\left( -U_0(x_0)+F_0 - \int_0^1 \nabla \cdot \hat b_t (X_t) dt\right)
> $$
> Therefore the weights are
> $$
> \frac{\rho_1(X_{t=1})}{\hat \rho_{t=1}(X_{t=1})}
> \equiv \exp\left( -U_1(X_{t=1})+ U_0(x_0)+F_1-F_0 - \int_0^1 \nabla \cdot \hat b_t (X_t) dt\right)
> $$
> On the other hand, if $\epsilon_t =0$, we have (using $\dot X_t = \hat b_t(X_t)$)
> $$
> -\partial_t U_t(X_t) -\nabla U_t(X_t) \cdot \hat b_t(X_t) = -\frac{d}{dt} U_t(X_t)
> $$
> so that the equation for the weights reduces to
> $$
> \frac{d}{dt} A_t = -\frac{d}{dt} U_t(X_t) - \nabla \cdot \hat b_t(X_t)
> $$
> and we deduce that
> $$
> A_{t=1} = - U_1(X_{t=1}) + U_0(x_0) - \int_0^1\nabla \cdot \hat b_t(X_t) dt
> $$
> Since $\mathbb{E}[e^{A_t}] = e^{-F_t+F_0}$, this shows that
> $$
> \frac{\rho_1(X_{t=1})}{\hat \rho_{t=1}(X_{t=1})} \equiv
> \frac{e^{A_{t=1}}}{\mathbb{E}[e^{A_{t=1}}]}
> $$
>
>
> [1] Pavliotis & Stuart, Multiscale Methods, 2008.

---

> > ### Author Response · Authors · 2024-12-03
> >
> > **Related work:**
> >
> > *CMCD:*
> >
> > We respectfully disagree that the relation to the PINN loss is given in Vargas Prop 3.3. Indeed, as you say the PINN loss is not even defined in their work. While it is clear you understand the nuances of the mathematics relating these claims, we do not think it is reasonable to claim that many of these relations can be easily intuited.
> >
> > Further, we are happy to cite this preprint that appeared two months before our submission, but the insistence of highlighting this work while also claiming that the insights of the PINN loss are already intuited in Vargas et al. seems contradictory and unfair. We hope that the reviewer can recognize this incongruity.
> >
> > We are happy to provide a citation to PDDS for also using SMC and could add its benchmarks to the corresponding tables for a camera-ready.
> >
> > We would like to push back on the notion that because you are able to relate the loss in Vargas et al to a control on the KL with your own mathematical insight that should limit the utility of our derivation of it for the PINN loss. This is bound is not in the literature, and while it nice to see this connection you have made, it should not hold bearing on the insight on how the PINN controls the KL. We hope that you can meet us eye to eye here.
> >
> > As for the action matching loss, we would like to re-stress that we have characterized the minimizer in a new form via Feynman-Kac, which is not in the action matching paper (See Appendix 4.3).
> >
> > *Discretization Errors:*
> >
> > That’s correct, the weights derived in CMCD also account for discrete time.
> >
> > *Log-Variance Loss:*
> >
> > We totally agree that the log-variance loss in theory and in practice for other settings can achieve strong performance. We are happy to mention this in related work. But please hear us out that we worked hand in hand with the CMCD authors to achieve the best results we could with the log-variance loss. In particular, the reported numbers are **with allowing them a learned time-dependent density, 256 sampling steps during learning (the method completely fails with fewer), and a fancier neural network, gradient clipping, etc.** We mean numerically unstable in the gentlest of terms and are truly happy to phrase this however the reviewer thinks is best. But unless we met these conditions, the method would not work, which we hope is reflective of the usefulness and novelty of our approach.
> >
> > We would in addition happily benchmark on more datasets to compare to CMCD, but there were no more readily available from the authors. We hope that this is not counted against us as we have already worked closely with them to get these results, which were non-trivial.
> >
> > We are happy to mention that other methods can be used on an arbitrary grid, and will simply stress that part of what we show here is that this can have significant positive effect, in that it allows us to a posteriori play with the $\epsilon \rightarrow \infty$ limit, which we have uniquely highlighted.
> >
> > *Experiments:*
> >
> > We did not receive a working codebase for the MoS from the authors, and only wanted to include their implementation to put CMCD in the best light.
> >
> > Please note that our cluster which housed the data to add the PINN to the phi4 plot was on maintenance during part of the short window of the rebuttal :slightly_smiling_face: But we have included the line for the ESS in an updated figure at this google drive link:
> > https://drive.google.com/file/d/1h_VvHyffzWXC4bvRDLsciK339YQsEyTp/view?usp=sharing .
> > As you can see, the performance as compared to action matching is very similar.
> >
> > Thank you for the reference on off-policy rhetoric, we can make these slight modifications shortly.
> >
> >
> > **In summary,** we really feel we have addressed many of your concerns, and have even done so in correspondence with the CMCD authors. We hope that you’d consider raising your score above a 3, as this is a stark score for a paper for which you have said “Generally speaking, I think that both the presentation and empirical evidence have improved.” Please let us know if there are any final adjustments we could make to change your mind, and we are happy to provide the slight amendments detailed above in a camera-ready submission.

---

> > > ### Comment · Reviewer_dGdu · 2024-12-03
> > >
> > > Thank you for the additional clarifications. Please note that I have not yet raised my score in the previous response since I wanted to wait for the remaining answers and explanations.
> > >
> > > **Question 1:** Thank you for the outline. Since this seems to be a promising and novel contribution, I would suggest including the details in a future version of the paper.
> > >
> > > **Question 2:** Thank you, it is clear now that this is a special case.
> > >
> > > **Related work (CMCD, PDDS, other PINN objectives, off-policy losses):**
> > >
> > > I did not intend to be "unfair" or "contradictory," and I apologize for any misunderstanding. My goal for mentioning these works during the review process was to improve the contribution of the paper by suggesting to (1) sufficiently discuss connections and differences to related works and (2) empirically compare against similar methods. While I think that the paper has improved in this direction (as mentioned in my review), I wanted to point out several topics that could still be further elaborated on.
> > >
> > >
> > > **CMCD, PINN loss, and KL control:**
> > >
> > > I want to emphasize again that I appreciate the different and arguably simpler derivations. I just think it is interesting to connect the results to previous work. Let me formulate the necessary steps to show how I see the connections to the PINN loss and the KL control:
> > >
> > > Proposition 3.3. in Vargas et al. (2024) directly (just renaming the variables according to your paper, i.e., $\hat{b}_t =\nabla \phi_t$ and $\ln \hat{\pi}_t = -U_t$, and using the fundamental theorem of calculus for $F_T-F_0$) yields that
> > >
> > > $$
> > > \log \operatorname{RND}(Y) =  \int_{0}^T \big( - \partial_t F_t + \partial_t U_t + \hat{b}_t \cdot  \nabla U_t - \nabla \cdot \hat{b}_t \big)(Y) \\, dt.
> > > $$
> > >
> > > Taking a process $Y$ with density $\hat{\rho}_t$ and using Jensen's inequality, we obtain that
> > >
> > > $$\mathbb{E}\left[\left(\log \operatorname{RND}(Y) + \int_0^T \partial_t F_t - \partial_t \hat{F}_t \\, dt\right)^2\right] \le L\_{PINN}^T[\hat{b}, \hat{F}].$$
> > >
> > > 1. *PINN objective*: If the PINN-Loss is zero, the log-RND must be almost surely constant (and thus zero), which implies that $\hat{b} = b$ (by the definition of the RND) and that $\hat{F} = F$ (since this argument can be made for any $T$).
> > > 2. *KL control*: By the data processing inequality and Jensen's inequality we have that $D_{KL}(\hat{\rho}_{t=1} || \rho_1) \le \sqrt{\mathbb{E}[(\log \operatorname{RND}(Y))^2]} \le  \sqrt{L\_{PINN}^T[\hat{b}, F]}$.
> > >
> > >
> > > **Experiments:**
> > >
> > > Thank you for the additional $\phi^4$ plot. While the paper would still significantly profit from further tasks/baselines, I understand that it is hard to obtain many additional experimental results in the short rebuttal period. However, I am not sure why additional tasks are not "readily available" since even the official codebase of CMCD (https://github.com/shreyaspadhy/CMCD) provides several other tasks.
> > >
> > > ---
> > >
> > > In summary, I will raise my score in the hope that the authors would sufficiently discuss related work and add additional experiments as written in their response.

---

> > > > ### Author Response · Authors · 2024-12-03
> > > >
> > > > We thank the reviewer for the clarifications, in particular how the PINN and the log RND interrelate in their KL control. We really appreciate your help in improving the paper.
> > > >
> > > > One final clarification for the experiments: working with the CMCD authors we had initial difficulty obtaining positive benchmarking on the repo you linked to. The CMCD authors graciously worked to translate their code into a different form that allowed for fairer/better benchmarking with our setup, but not all benchmarks were translated over. We continue to correspond with them to add additional experiments.
> > > >
> > > > We will happily add these additional benchmarks and clarifications. We appreciate your increased rating.

---

### Author Response · Authors · 2024-11-25
**General Reply to Reviewers**

We thank the reviewers for their careful reading of our paper and for their constructive comments that helped us  improve our results as well as our presentation. *The main changes and addititions made in the revised version are marked in orange.*

We reply to each reviewer separately below but we first address some common concerns in this general response, highlighting some differentiating factors of our work from previous literature:

**Novelty and utility of the PINN loss:** We would like to pushback on the notion that the utility of the PINN loss in this context was already well established. We thank you for pointing to some of these citations, particularly arxiv.2301.07388, on other instantiations of it, and have accordingly adapted this into the writeup. However, these works do not make the following important points, which we establish:
- **Validity of the PINN in the context of annealed langevin dynamics.** In prior works it was not not shown that the PINN loss (which is independent of $\epsilon_t$) could be used in the context of annealed Langevin dynamics with *any* $\epsilon_t$.
- **New insights interpreting the PINN loss as a control on the variance of the Jarzynski weights.**
- **New insights  showing that the PINN loss controls the KL divergence.**

**Realizability of the $\epsilon_t \rightarrow \infty$ limit of annealed Langevin dynamics with transport.** Working at the level of the PDE, it is easy to see that perfect sampling is achieved in this limit, whether the learned transport is perfect or not. In general, this limit cannot be reached in practice without transport (as it would require taking astronomically large value of $\epsilon_t$ in general), but we show that, with some learned transport added, *even moderate values of $\epsilon_t$ can improve the sampling dramatically*. This feature can be exploited after training as an *explicit knob for tuning performance vs cost*. This has not been recognized in other ML sampling literature, a fact that the authors of CMCD suggested to stress in this response. The improvements that can be achieved by playing with $\epsilon_t$  highlighted Figure 2, and we have also added a simple plot to the appendix about what this knob gives in terms of Wasserstein-2 distance on a new Mixture of Student-T example (50-dimensional problem).

**Incorporation SMC resampling into the generation process.** We point out that the Jarzynski weights can be used on the fly during generation to perform resampling in the style of sequential Monte-Carlo. We show that this can increase the performance of the method.



**New experiments and re-treatment of existing ones:**
- We have corresponded with the authors of both CMCD (Vargas et al 2024) and the Beyond ELBOs (Blessing et al 2024) paper to directly implement their benchmarking protocols for  new and exisiting experiments and compare to an existing CMCD implementation for the Gaussian mixture example on which it was already done. This is to construct an apples to apples comparison of the methods.
- We have readjusted how we evaluate our method using the evaluation sizes etc as asked by reviewer dGdu. Every evaluation now uses 100 sampling steps. For comparison on the GMM, because we quoted results from the iDeM paper, we use their W2 benchmark on their quoted amount of samples (1000). For all other experiments, we quote Blessing et al. Please note that we initially only trained our models for the smaller problems for 2500 training steps. Allowing them to go to 10k, training steps, we see vast improvement in performance.
- We have added an additional higher dimensional experiment on the 50-dimensional Mixture of Student-T distribution from the Beyond ELBOs paper.
- We have now tested the PINN loss on the lattice field theory examples and observe nearly equivalent performance of the  method as compared to action matching. We have done this to provide results to show that both work in high dimensions.

If you are satisfied with these adjustments, we would appreciate any improvement in your score. Thanks again for your valuable feedback that has improved the paper.

---

### Meta-Review · Area_Chair_qXGq · 2024-12-18

**Metareview:**

This study proposes a novel sampling method inspired by insights from nonequilibrium physics to enable efficient sampling from unnormalized distributions. The method employs stochastic differential equations (SDEs) and introduces auxiliary dimensions to reformulate the process within a new framework of annealed importance sampling (AIS). Ultimately, an additional drift term is incorporated to mitigate the impact of weight variance in AIS, resulting in a new sampling approach. Considering the quality of the reviews, I (AC) had carefully examined the manuscript. While the research direction is interesting and important, significant concerns remain regarding the presentation, mathematical rigor, and the numerous typographical errors, which cast doubts on the manuscript's accuracy. For these reasons, I recommend rejection at this time. Below are the specific points of concern:
- The authors frequently rely on the relationship between SDEs and the PDEs (FP equations) without providing proofs or referencing existing literature. While the kinetic FP equation seems to be corresponding their approach, many of the PDEs appear to be proposed newly in this paper and thus, a formal discussion about the  derivation is necessary. Additionally, there is no discussion of the functional space in which these PDEs are defined. For instance, it is unclear whether Equation (36) does not diverge. Since these equations are used to ensure the mathematical rigor of the sampling process, such discussions are essential.

- In the proof of Proposition 1 in Section 4.1, the authors use integration by parts (e.g., at the end of Equation (37)). Since this holds in under moderate assumptions, the authors should include a discussion about it, such as the support of the distribution and the behavior of the density function at its boundaries to justify this step.

- The paper contains many typos, as pointed out by all reviewers, significantly impacting readability. Even after the discussion, in Equation (19), the left-hand side's $\nabla U_t$ should be $\nabla U_t \cdot \hat{b}_t$. Similar copy-paste errors are found in multiple other parts of the manuscript.
- In Proposition 2, $T$ is not defined.
- The proof of Equation (9) is claimed to be in Appendix 4.1, but no such proof exists. While Equation (41) in the appendix appears to be related, it does not constitute a complete proof.

**Additional Comments On Reviewer Discussion:**

Concerns regarding the theoretical properties of the proposed method and its comparisons with existing approaches were raised. For example, Reviewer XBnC and Reviewer dGdu highlighted issues related to discretization errors, leading to the addition of new theoretical analysis in Section 4.2 to address these points. Additionally, new comparative experiments were conducted, including an application to the $\phi^4$ model. However, these efforts were still deemed insufficient by reviewers.
Many reviewers also pointed out the large number of typographical errors. While efforts were made to address these issues, typos and omissions remain in the revised manuscript. This serves as significant evidence that the paper has not yet reached a publishable standard.

---

### Decision · Program_Chairs · 2025-01-22

Reject